# Versatile live-cell activity analysis platform for characterization of neuronal dynamics at single-cell and network level

Xinyue Yuan [1]✉, Manuel Schröter[1], Marie Engelene J. Obien[1,2], Michele Fiscella[1,2], Wei Gong[1,2], Tetsuhiro Kikuchi [3], Aoi Odawara[4], Shuhei Noji[4], Ikuro Suzuki[4], Jun Takahashi[3], Andreas Hierlemann [1,5] & Urs Frey [1,2,5]

Chronic imaging of neuronal networks in vitro has provided fundamental insights into mechanisms underlying neuronal function. Current labeling and optical imaging methods, however, cannot be used for continuous and long-term recordings of the dynamics and evolution of neuronal networks, as fluorescent indicators can cause phototoxicity. Here, we introduce a versatile platform for label-free, comprehensive and detailed electrophysiological live-cell imaging of various neurogenic cells and tissues over extended time scales. We report on a dual-mode high-density microelectrode array, which can simultaneously record in (i) full-frame mode with 19,584 recording sites and (ii) high-signal-to-noise mode with 246 channels. We set out to demonstrate the capabilities of this platform with recordings from primary and iPSC-derived neuronal cultures and tissue preparations over several weeks, providing detailed morpho-electrical phenotypic parameters at subcellular, cellular and network level. Moreover, we develop reliable analysis tools, which drastically increase the throughput to infer axonal morphology and conduction speed.

[1] Department of Biosystems Science and Engineering, ETH Zurich, Basel, Switzerland. [2] MaxWell Biosystems AG, Zurich, Switzerland. [3] Center for iPS Cell Research and Application (CiRA), Kyoto University, Kyoto, Japan. [4] Tohoku Institute of Technology, Sendai, Japan. [5]These authors jointly supervised this work: Andreas Hierlemann, Urs Frey. ✉email: xinyue.yuan@bsse.ethz.ch

Recent advances in brain imaging have provided detailed new insights into the exquisite complexity of the human brain[1–3]. However, translating this knowledge into effective treatments for severe neurodevelopmental and neurodegenerative diseases remains challenging. Traditional electrophysiology studies using rodent samples, such as primary neuronal cultures and acute/organotypic preparations (e.g., brain slices, retina, etc.), have been essential to understand the basic brain functions and neuronal dynamics[4–6]. Although, validating the applicability of results from animal models to human conditions proves to be difficult. The advent of induced pluripotent stem cells (iPSCs) has provided an attractive avenue to study aspects of some brain disorders in living human tissue in vitro. As they retain the genetic signatures of their donors, fibroblasts, which are extracted from healthy subjects or patients, can be reprogrammed and differentiated into neurons[7,8]. The use of human cells as disease models offers a promising approach for functional phenotyping[9,10], high-throughput drug-screens, and dissecting neuronal function using pharmacological or genetic challenges[11,12]. Currently, an approach for comprehensive functional characterization of neuronal cultures is still lacking, as it requires a high-throughput method with high spatial-temporal resolution to capture neuronal features at subcellular, single-cell, and network levels.

Ideally, live-cell functional imaging permits (1) continuous and simultaneous measurements from cells, or populations of cells, without perturbation of the sample. Moreover, live-cell functional imaging should be (2) highly sensitive and offer sufficient spatiotemporal resolution to detect biologically relevant details that unfold over days and weeks. Finally, it should allow for (3) reliable data acquisition and (4) feature extraction that ensure high reproducibility of the results. Existing optical imaging techniques focus mainly on neurite growth, morphology, cell viability, and functional activity, but cannot cover all these factors at the same time. Moreover, most imaging methods require labeling of cells by fluorescent indicators[13,14]. Live-cell-imaging techniques exist[15], but still are limited in their temporal resolution and prevailingly rely on fluorescence markers[14,16], the use of which can entail phototoxicity and alter the physiology of the cells[17,18].

The goal of this study is to develop a technology for comprehensive and effective electrical live-cell functional imaging and phenotypic screening, which we foresee will facilitate the detailed study of a wide range of preparations and cells, including primary and iPSC-derived neuronal networks. Today's electrophysiology methods range from classic tools, such as patch-clamping, which can provide the detailed recordings of membrane potentials or even single-ion-channel activity of patched individual cells, to highly parallel network activity recordings with neural probes featuring a multitude of channels. Recently introduced high-density microelectrode arrays (HD-MEAs), based on complementary metal-oxide-semiconductor (CMOS) technology, have enabled simultaneous access to the electrical activity of thousands of neurons at high temporal and spatial resolution[19–27], leading to what can be termed "electrical functional imaging". Compared to traditional optical imaging, such as light microscopy, electrical functional imaging by means of HD-MEAs captures the electrical activity of neurons, such as extracellular action potentials (APs) and local-field potentials (LFPs). These signals can then be used to assess neuronal function and neural network dynamics. This type of imaging has been enabled by the development of different HD-MEA circuit architectures[19], each with certain strengths and weaknesses. Full-frame architectures with so-called active-pixel-sensors (APSs)[20–23], enable simultaneous recording from every electrode of the array, but are limited by relatively low signal-to-noise ratios (SNRs) caused by circuit-design constraints, i.e., the little available area in a pixel to realize high-performance circuits. This architecture is therefore suited mostly for recording of large-amplitude signals, such as APs of mature primary neurons and LFPs. Another HD-MEA-based method involves recording from the selected areas or subsets of electrodes of the array at a given time at high SNR. The circuitry architecture used for this approach is called switch-matrix (SM), which includes a flexible switching and addressing scheme within the array. The front-end amplifiers are placed outside of the array area, where there is sufficient area to implement large low-noise amplifiers[24–27]. The SM approach permits the detection of signals with high sensitivity at dense electrode packing. Due to the high SNR, it also enables the detection of small signals, such as axonal signals and APs of immature neurons, e.g., from iPSC-derived cells. However, recording with SM-based HD-MEAs can lead to suboptimal characterization, as only signals from a subset of electrodes are captured at a time, and as the scanning of all electrodes to carry out network studies requires considerable time.

To address some of the limitations of currently available HD-MEAs, we develop a device that combines APS and SM capabilities for live-cell electrical functional imaging—the dual-mode MEA (DM-MEA)[28]. This CMOS-based HD-MEA (Fig. 1a) features an array of 19,584 electrodes at a pitch of 18.0 μm within an area of $1.8 \times 3.5$ mm$^2$ and an electrode density of 3050 electrodes/mm$^2$. A schematic of the DM-MEA is shown in Fig. 1b. While all electrodes can be continually read out using the APS mode, a subset of 246 electrodes can be simultaneously read out using the SM mode. The full-frame readout (APS mode) features a noise level of 10.4 μV$_{rms}$ at 11.6 kframes/s, while the 246 readout channels of the SM mode have low noise levels of 3.0 μV$_{rms}$ at a sampling rate of 24.4 kHz. These low-noise readout channels can be flexibly connected to an arbitrarily selectable subset of the 19,584 electrodes in the array using the SM routing scheme[26]. Compared to previous MEA architectures and systems[20–27], the DM-MEA system provides large-scale parallel data acquisition from the full array at high electrode density, while offering the option of simultaneous high-SNR readout of areas of interest. Despite the circuit design challenges to integrate two modes into one device, the full-frame read-out circuitry has sufficient SNR to reliably record signals with amplitudes of approximately 50 μV (five times RMS noise). This SNR specification represents a significant improvement over current state-of-the-art systems[19] with an electrode density of 3050 electrodes/mm$^2$. Using the SM-mode feature of the system, signals with amplitudes as low as 15 μV can be steadily detected.

The combination of the two recording modes substantially reduces the data acquisition time, rendering the system suitable for preparations that exhibit small signals. For example, spike times and trigger points could be extracted from high-SNR SM-mode recordings, which then could be used for averaging of signals, captured in full-frame APS mode, so that complete "electrical images" could be reconstructed. Concurrently, we have developed the efficient methods that facilitated accurate quantification and robust statistical analysis of the acquired large-scale electrophysiological data to extract single-cell, subcellular and network dynamics of hundreds of neurons from recordings lasting only a few minutes.

We demonstrate the broad applicability of the system in different in vitro preparations (Figs. 2, 3) and focus on three functional assays that can be performed with the DM-MEA (see also Fig. 1c): (1) The whole-sample-activity-imaging assay produces dynamic videos of the spontaneous activity of the neurons in the sample in addition to providing information about the location and movement of the cells during culture development. (2) The axonal-arbor assay provides a proxy-readout for the morphology of axons and the conduction velocity along each branch[29,30]. Compared to assays using high-content imaging techniques[31], axonal morphology and functional changes in individual neurons can be tracked in this work. While electrical imaging of axonal

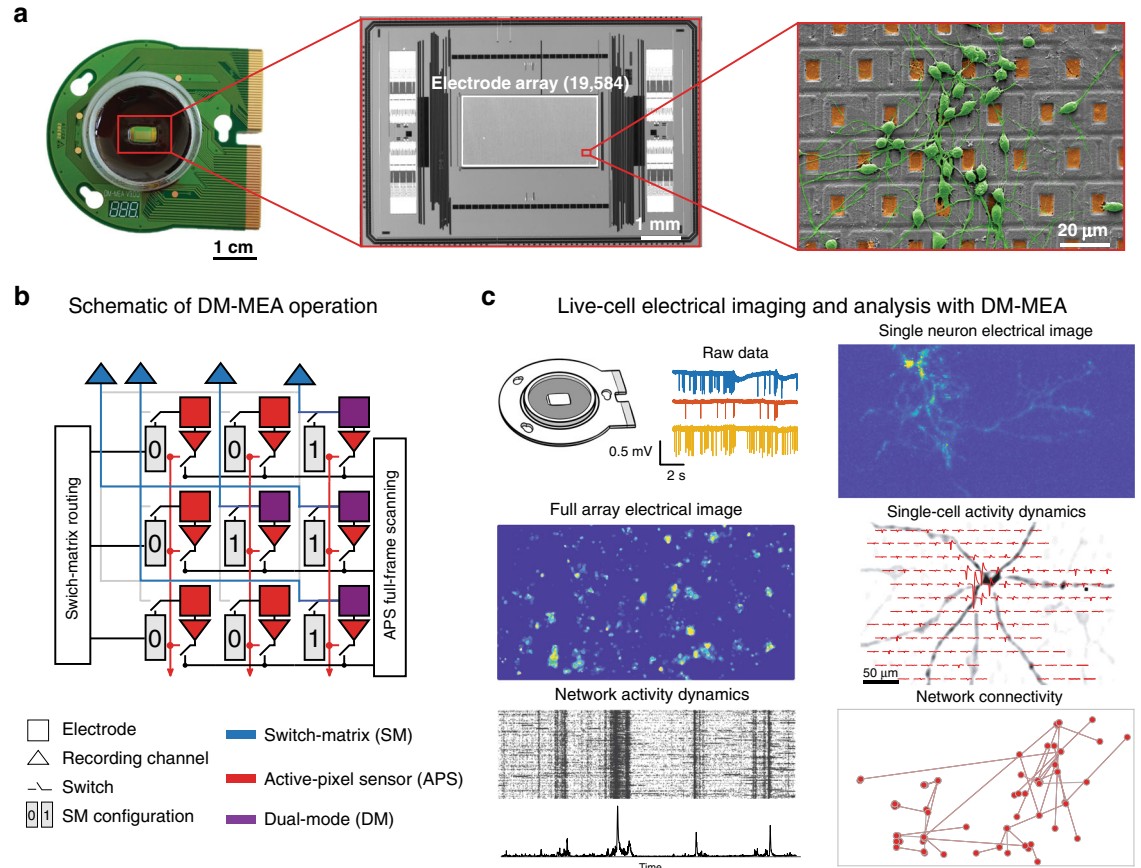

**Fig. 1 Dual-mode microelectrode array (DM-MEA) system, applications and readouts. a** Overview of the HD-MEA system, including the packaged device used for biological experiments, a micrograph of the CMOS DM-MEA and a scanning-electron micrograph (SEM) of primary neurons cultured on top of the DM-MEA. **b** Schematic concept of the dual-mode readout. Every electrode is continuously read out in the APS mode (red), while a few selected electrodes are simultaneously read out in the switch-matrix mode (purple). **c** Experimental work flow using the DM-MEA, including acquisition of electrophysiological signals with different features, and data analysis to extract various features.

arbors has been demonstrated in previous work[29,30], the number of axonal arbors that could be studied was limited to tens of cells per sample. The method presented here enables us to drastically increase the throughput to hundreds of cells per sample, a performance improvement that is required for large-scale screening applications. Finally, the network-connectivity assay (3), which provides the extent to which each neuron is synaptically connected to the rest of the network, i.e., the number of putative connections in a sample and information on the signal propagation and synaptic delays between neurons. In summary, these assays can provide a comprehensive overview on the functional dynamics of cells, their maturation over development and phenotypic characteristics, as well as their response to pharmacological tests.

## Results

**Combining full-frame and switch-matrix mode recordings.** By combining the two recording modes, full-frame active pixel sensor (APS) and high-SNR switch-matrix (SM) mode, the DM-MEA can significantly enhance the characterization of extracellular signals of neurogenic cells. Figure 2a illustrates the concept behind DM-MEA recordings through schematic images: the extracellular activity of neurons comprises a mixture of large-amplitude signals near the soma or the axon-initial segment (AIS) and small signals along the axons. Recordings in full-frame APS mode can be used to detect signals featuring larger amplitudes over a large area (i.e., the whole

array), whereas recordings in SM mode can be used to capture more details and signals with small amplitudes in user-defined regions of interest, so that low-level electrophysiological signals that are invisible to the APS mode can also be detected. This feature is of particular interest for iPSC-derived neurons, as their signal levels are often so low that even signals near the soma or AIS are not detectable using the APS mode alone. Here, the experimenter can, for example, use the high-resolution signals, recorded simultaneously from electrode subsets in SM mode, to extract spike times and trigger points that can be used to average the signals captured in APS mode (so-called spike-triggered averaging (STA))[29,30]. Applying such recording strategies, DM-MEAs can efficiently reconstruct a full electrical image.

Here, we demonstrate the potential of DM recordings with sets of signals recorded from rat primary cortical neurons and human iPSC-derived glutamatergic neurons. The topmost two signal traces in Fig. 2b show extracellular signals of a primary cortical neuron recorded simultaneously in APS and SM mode through the same electrode. Recordings were performed at day in vitro (DIV) 20. The neuron featured large-amplitude spikes clearly exceeding 100 μV (Fig. 2b). APs could be easily detected and used to further study the extracellular-waveform distribution of the neuron on the array (electrical footprint). Detected APs were used as trigger points for spike-triggered averaging of the signal of neighboring electrodes. In Fig. 2b, we used the spike times of the top neuronal signal trace as triggers to cut out sections of the corresponding raw data (±2 ms

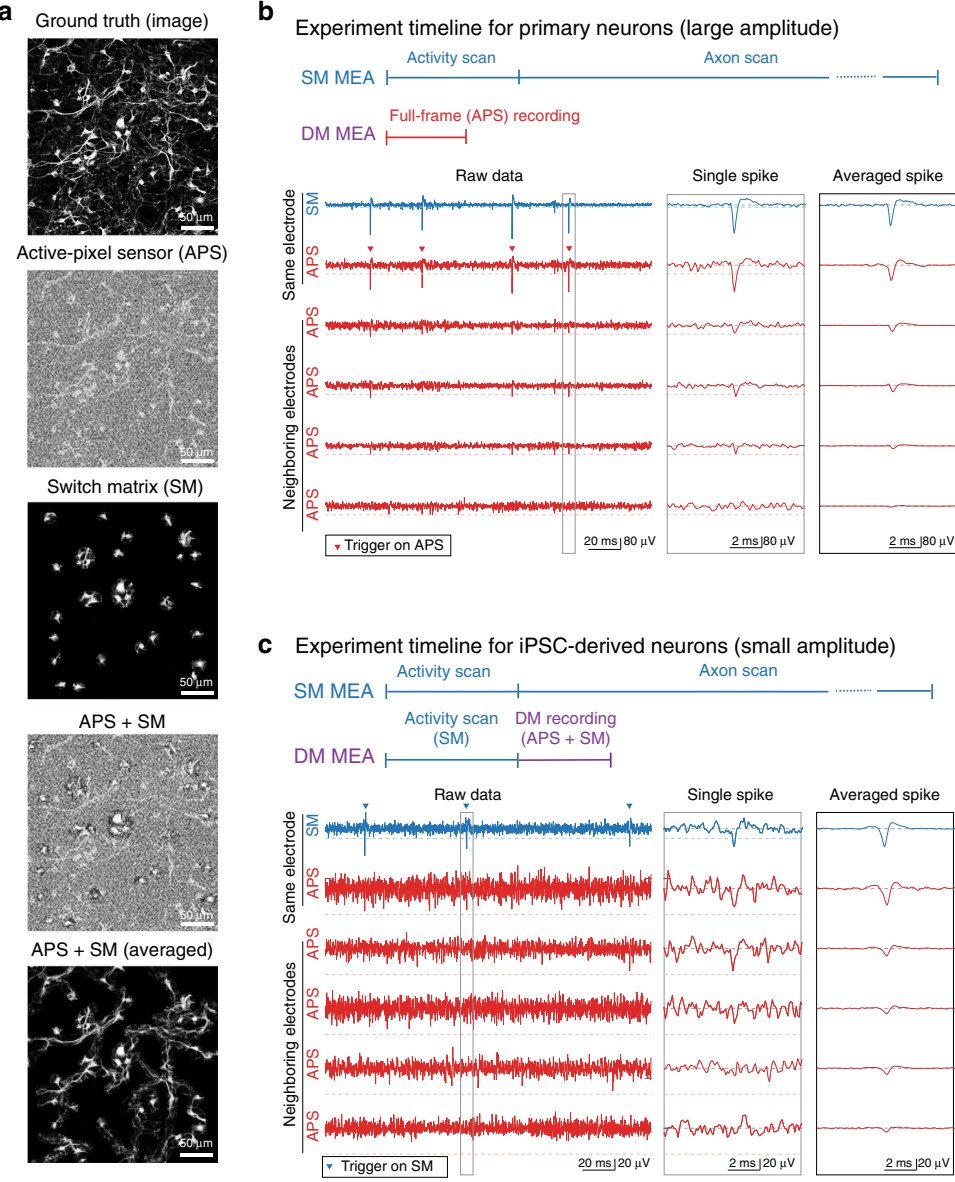

**Fig. 2 Key concept of SM, APS, and combined readouts. a** Illustration of the DM concept, using optical images of primary neurons instead of electrical recording to outline the concept. **b** Experimental workflow for primary neurons (including comparison to SM-only HD-MEAs), and simultaneous recording by using APS and SM modes. Primary neurons provide clearly visible spikes in the raw data acquired in both modes. The electrical footprints can be extracted by spike-triggered averaging using either of the modes. The topmost two signal traces in SM and APS mode have been recorded from the same electrode featuring the largest signal amplitude. Both modes provide large enough signals and clear trigger points (filled triangles). **c** Experimental workflow for iPSC-derived neurons (including comparison to SM-only HD-MEAs), and simultaneous recording by using APS and SM modes. IPSC-derived neurons typically yield rather small-amplitude spikes, which can be detected in the SM data but not in the APS data. The electrical footprints can then be extracted by spike-triggered averaging of APS data using the SM mode and the spikes extracted from the SM mode as trigger signals. Again, the topmost two signal traces in SM and APS mode have been recorded from the same electrode featuring the largest signal amplitude. Only the SM mode provides a large enough signal-to-noise ratio and clear trigger points (filled triangles).

before/after the AP spike). Averaging over multiple spike events reduced the influence of noise and revealed small signals originating from the axonal/dendritic compartments of the cell. For mature primary cortical neurons featuring large-amplitude spikes, spike-triggered averaging worked equally well for extracting the trigger points from recordings in SM or APS mode (both topmost traces in Fig. 2b show clearly visible spikes). The "electrical footprint" of the neuron across a large number of electrodes could be obtained after spike-triggered averaging. The ability to record full-frame data substantially shortened and simplified the experiments in

comparison to only using the SM, where many configurations of electrode subsets would have to be sequentially applied. Sequential activity scans to find active electrodes and to reconstruct axonal signals require >30 min per chip with SM-only chips[29,30], whereas a 3-min recording in APS mode provided comparable results for this specific preparation.

For iPSC-derived neurons, however, even the largest signal amplitudes of spontaneous spikes often amount to less than 50 μV and were, thus, difficult to detect with the APS readout, while the use of the SM mode still revealed clear signals (see two top traces in

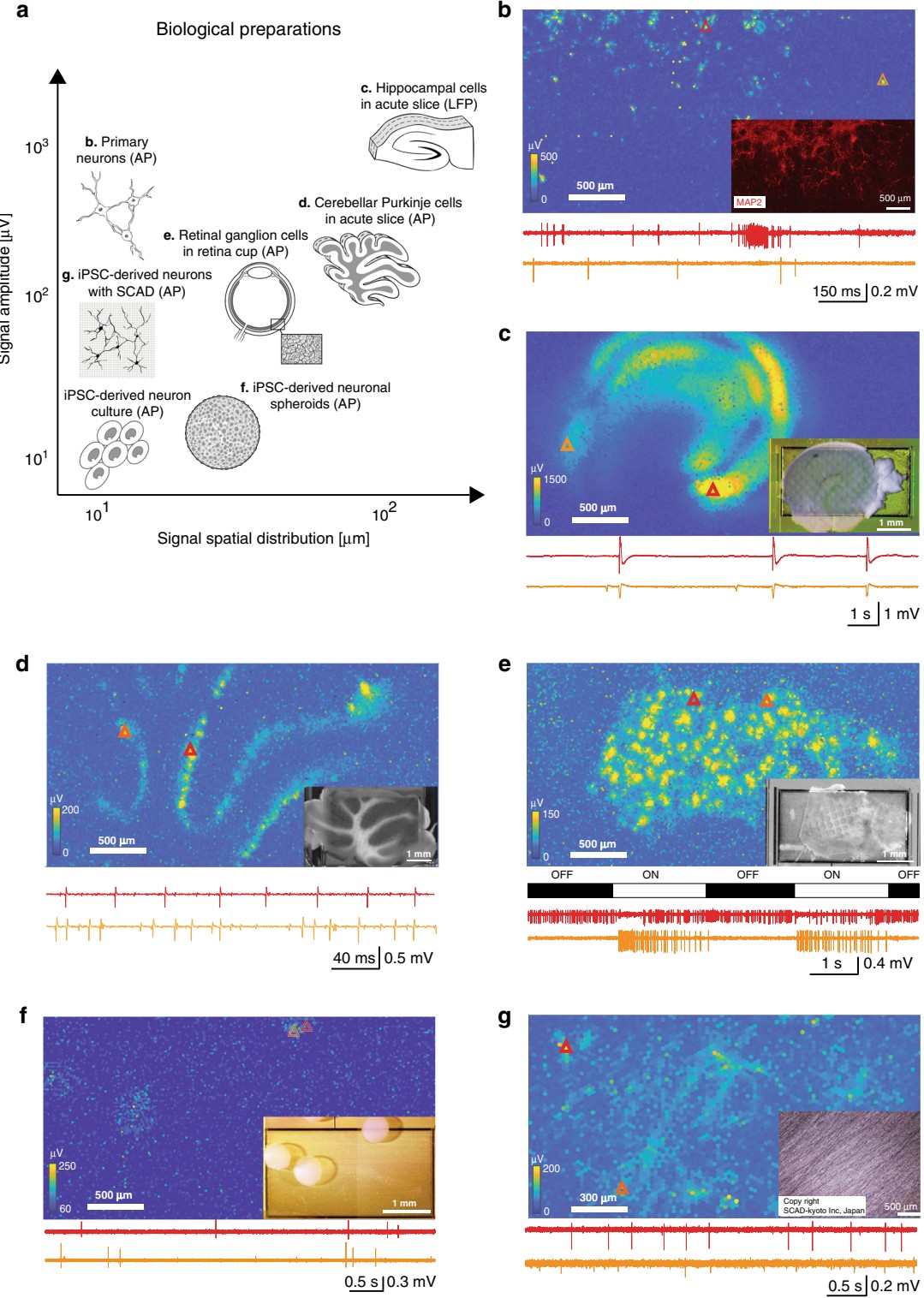

**Fig. 3 Different biological preparations: optical images, electrical-activity maps, and example raw traces.** The red and orange triangles mark the positions, where in each preparation the two exemplary signal traces have been recorded. **a** Summary of biological preparations shown in **b**–**g**, featuring different signal source spatial distributions and signal amplitudes (courtesy of MaxWell Biosystems AG). **b** Action-potential (AP) recording from primary rodent neurons. **c** Local-field-potential (LFP) recording from acute hippocampal slices. **d** AP recording from acute cerebellar slices. **e** Recording from an acute retina preparation including light responses. **f** Recording from iPSC-derived neuronal spheroids. **g** Recording from iPSC-derived neurons using a SCAD scaffold device for 3D cell cultures (SCAD Kyoto Inc). For each preparation, more than three independent experiments (technical repetitions) were conducted, yielding comparable results.

Fig. 2c). Still, the full electrical footprint of the neuron could be recovered by simultaneously recording in SM and APS mode: we used the spike times detected in the SM recording mode as trigger points (Fig. 2c, top blue trace) for the spike-triggered averaging of the APS recordings. Since the APS mode recordings covered the full array, the spatial profile of the neuron not only became visible near the trigger electrode, but also across the entire array, which allowed for reconstruction of complete neuronal footprints. Here, the workflow using the DM-MEA also included sequential short-time activity scans across all array electrodes in SM mode to find the active electrodes. Nevertheless, the overall measurement procedure was substantially shortened by the use of the DM mode (APS + SM), as the axonal signals could be obtained through averaging in the APS mode and time-consuming, sequential scans of electrode subsets could be avoided.

**Whole-sample activity mapping of various biological preparations**. We recorded the electrical activity of various biological preparations using all array electrodes. Figure 3 demonstrates the broad applicability of the DM-MEA for live-cell activity imaging using full-frame APS mode, including recordings from primary rodent neurons, brain slices, retina, and 3D neuronal spheres.

*Primary neurons*: Dissociated primary rat neurons were plated and cultured on the DM-MEA at a cell density of 3000 cells/mm$^2$ as described previously[30]. Recordings started approximately one week after plating. Figure 3a and Supplementary Movie 1 show the spontaneous electrical activity of developing neurons across the entire array, which appear similar to recordings of spontaneous electrical activity obtained through wide-field optical voltage imaging[32]. It should be noted, however, that the present recordings have a significantly higher temporal resolution and larger recording area. Propagating axonal signals are clearly visible in the raw data shown in the movie for many of the neurons. To the best of our knowledge, such distinct spontaneously propagating action potentials of multiple neurons, detected simultaneously over large areas by HD-MEAs, have not been shown previously. Owing to the high spatiotemporal resolution, less averaging was required to extract AP propagations along axonal arbors of individual neurons.

*Hippocampal slices*: Seizure-like LFP events were obtained from 300-μm-thick acute mouse hippocampal slices (Fig. 3b and Supplementary Movie 2; band-pass filtered signals: 1–300 Hz). The DM-MEA recordings clearly indicated different anatomical subregions. Seizure-like LFPs, induced by 4-Aminopyridine (4-AP) application, propagated from the CA3 to CA1 region, potentially indicating functional connectivity between the two regions[33] (Fig. 3b). Compared to previous work using HD-MEAs for LFP recordings[34], the larger sensing area of the DM-MEA enabled activity mapping of the full hippocampal slice at higher spatiotemporal resolution (five-times the number of electrodes).

*Cerebellar slices*: We also recorded activity from acute cerebellar slices of 3-week-old mice (Fig. 3c; band-pass filtered signal: 0.3–5 kHz). Cerebellar Purkinje cells, which become GABAergic upon maturation, likely represented the main source of the extracellularly detected APs here. Using the DM-MEA, we recorded from the full brain slice simultaneously including all Purkinje cells, which has not yet been shown with other HD-MEAs[6,35] at such high spatial resolution. Moreover, we could map the Purkinje cell layer and cerebellar lobules across the full brain slice at high spatiotemporal resolution.

*Retina*: Different types of retinal ganglion cells (RGCs) can be identified in ex vivo retinae using HD-MEAs[36]. We imaged APs of single RGCs by placing an excised piece of a rat retina on top of the DM array and extracted functional features and axonal traces from hundreds of RGCs (Supplementary Fig. 1 and Supplementary Movie 3). The measured baseline firing rate was 13.5 Hz, and most of the RGCs featured only one axon projecting in the direction of the optical nerve. We stimulated the RGCs with full-field ON–OFF light stimuli and obtained ON and OFF RGC light-triggered responses (Fig. 3d). The extraction of axonal morphologies and AP-propagation velocities of hundreds of RGCs has not been previously shown at comparable spatial resolution. The data in Supplementary Movie 3 demonstrates the system performance in retinal applications, which is a consequence of combining a large array, high spatial resolution (improved spike sorting performance[36,37]), and the comparably low noise of the APS readout.

*Neuronal spheroids*: In addition to 2D neuronal cultures, we also measured APs of 3D spheroids of dopaminergic-neurons that were derived from human iPSCs[38]. The spheroids were attached to the electrode array, which was coated with polyethylenimine (PEI) (Fig. 3e). The signals were most likely obtained from the neurons at the outer surface of the spheroid. In the activity map, the positions of the spheroids were evident. The large available sensor area allowed for simultaneously recording from multiple spheroids.

*SCAD*: A SCAD device (SCAD, Stem Cell & Device Laboratory, Inc., Kyoto, Japan) is a nanofiber scaffold that can be used to attach neuronal cells to build 3D cell cultures[39]. By placing iPSC-derived human dopaminergic-neurons (Elixirgen Scientific, Baltimore, USA) that have been cultured on an SCAD device on the electrode array, we recorded APs from 3D tissue-like neuron cultures (3–4 cell layers, 50 μm thickness), which showed features that were different from those observed in 2D neuronal cultures (Fig. 3f). Moreover, we observed signals propagating along the membrane structure (Supplementary Movie 4), likely indicating APs traveling along axonal arbors. These propagating signals featured higher amplitudes (>100 μV) as compared to axonal signals in 2D iPSC-derived neuron cultures shown in Fig. 2c (signals amplitudes <50 μV). This larger signal amplitude may be a consequence of the membrane-like structure of the neuronal layers on the SCAD, which may act as an insulating layer on top of the electrode array and, therefore, help to amplify the signals[40]. The detection of propagating axonal signals over extended areas in raw HD-MEA data, without any averaging, has, to the best of our knowledge, not been demonstrated previously.

**Fast and reliable extraction of subcellular neuronal features**. Following whole-sample activity imaging, we extracted several subcellular neuronal features from DM-MEA recordings. As exemplary preparation, we used mature primary neuron cultures (DIV 20; Fig. 3a). As illustrated in Fig. 2, the signals of primary neurons feature a comparably large amplitude, which could be detected using the APS mode alone. Therefore, we used APS full-frame readout data in the following analysis.

After a semiautomatic spike sorting procedure (see "Methods" section), we performed spike-triggered averaging and inferred the neuronal footprints (Supplementary Movie 5). Based on the signal flow of propagating APs (Fig. 4), STA enabled the reconstruction of full axonal arbors. The morphology of the axonal arbors has been validated through simultaneous use of HD-MEAs and optical imaging techniques[29,30,41]. STA enabled a quantification of a range of morphoelectrical properties of the neuronal footprints (Fig. 5). Specifically, we determined, in addition to the spiking activity metrics, (i) the AP propagation velocity along different axon branches and extracted different morphological features related to the axons of single neurons, including (ii) the longest axonal segments and (iii) the total extension of the axonal arbor on the DM-MEA.

Figure 5a depicts the activity and latency maps of three selected neurons. To calculate the velocity for each axonal segment, we

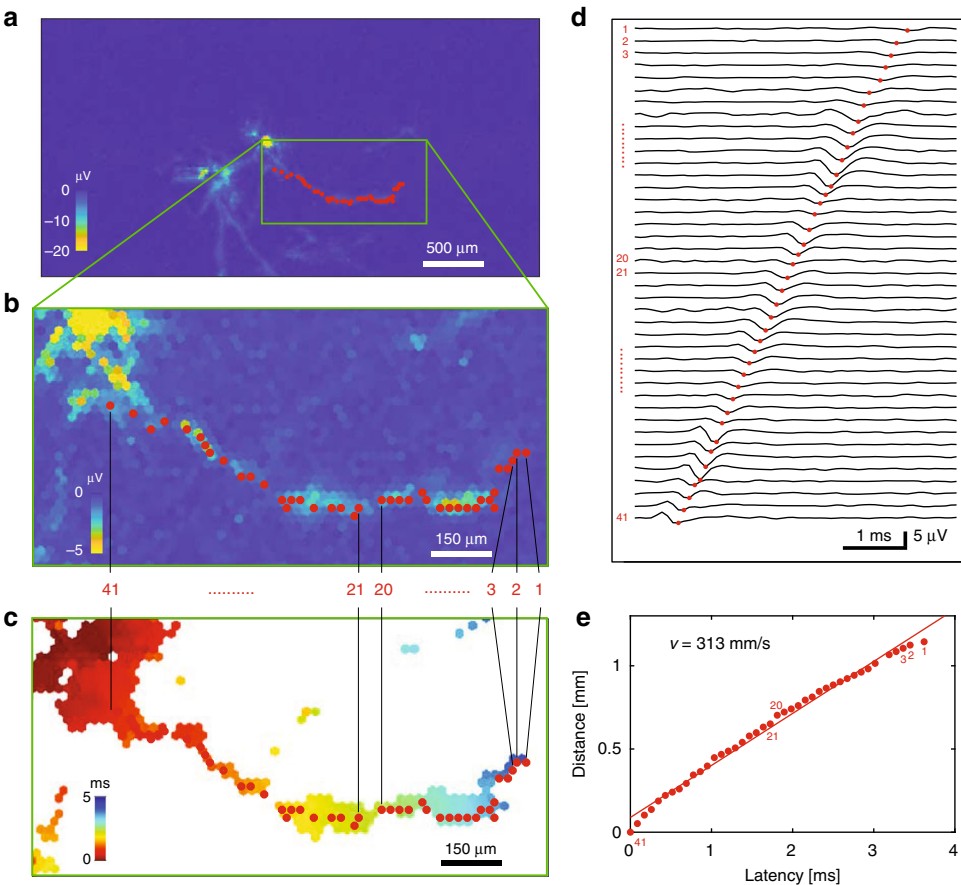

**Fig. 4 Extraction of axonal segments. a** Footprint of one neuron (signal amplitude is color-coded) with selected axonal segment marked with red dots; **b** zoom-in view of the selected axonal segment (signal amplitude); **c** delay map of the electrodes in the close-up view in **b**; **d** waveforms from the selected electrodes in **b** and **c**; and **e** linear fit of distance versus latency for the selected axonal segment in **b**–**d**. The extraction procedure started with the last frame $N$ featuring the largest delay (marked with 1), and proceeded backwards in time along the axonal structure. The next point was chosen by selecting the electrodes featuring a spike in the frame $N-1$ within a radius of 80 μm from the preceding point. If there were multiple electrodes within the same frame (e.g., between electrode numbers 20 and 21), the electrode with the largest amplitude was chosen. If there was no valid electrode within the frame $N-1$, electrodes within the frame $N-2$ were chosen according to the same procedure. The extraction was stopped five frames before reaching the signal initiation region and the corresponding first frame (e.g., at point 41 in the figure) in order to avoid including the signals of the electrical-activity initiation site (typically the distal end of the axon initial segment (AIS)). The signals, generated by the AIS, are very large and visible on many electrodes, so that precise signal timings are difficult to assess in that area.

first extracted, for each electrode, the latency of the recorded axonal signal as well as the distance from the signal initiation site (AIS location); the velocity was then estimated by applying a linear regression[30]. Details about the extraction method, which always started at the end of the axonal branches and segments and tracked the signal backwards towards the initiation site (AIS), can be found in the "Methods" section and are displayed in Fig. 4.

After the extraction of axonal segments, we were able to extract information about the morphology of neurons, including the length of the longest segments; more importantly, by adding up all detected segments of a neuron, we could get an estimate of the total extension of the axonal arbors, which could be used to estimate the outgrowth of neurites and axons.

Figure 5b summarizes the results extracted from 1086 spike-sorted neurons (pooled over five chips recorded on DIV 20). We found for each neuron on average $4.1 \pm 3.1$ axonal segments; the average over all longest extracted axonal branches amounted to $780 \pm 450$ μm. We then estimated the total extension of the axonal arbors by adding up all segments (total of approximately 4500 segments). The average axonal arbor extension over all neurons was $2200 \pm 2000$ μm. The electrical-signal propagation velocity in the axonal segments,

averaged over all neurons, amounted to $480 \pm 90$ mm/s, which is consistent with that found in previous studies[29]. The neuronal firing rate showed an asymmetric distribution around an average value of $2.4 \pm 2.4$ Hz. The average maximum signal amplitude (negative peak at AIS) within the inferred electrical footprints of the neurons amounted to $150 \pm 75$ μV.

**Mapping network activity and putative monosynaptic connections.** Next, we used the DM-MEA recordings to study burst dynamics and functional connectivity of developing neuronal cultures. All cultures showed spontaneous activity with occasional network bursts, as previously reported for developing neuronal networks[42]. Figure 6a shows the spike-sorted network activity of a rodent primary-neuron culture (DIV 20, $n = 204$ neurons); Fig. 5b summarizes the burst durations and interburst intervals (IBI) over five cultures ($n = 1086$ neurons). We found a burst duration of $0.8 \pm 0.3$ s, with an IBI of $5.7 \pm 3.6$ s, which were largely in line with the characteristics that have been reported in the literature[42]. However, we could record with much higher throughput from each sample.

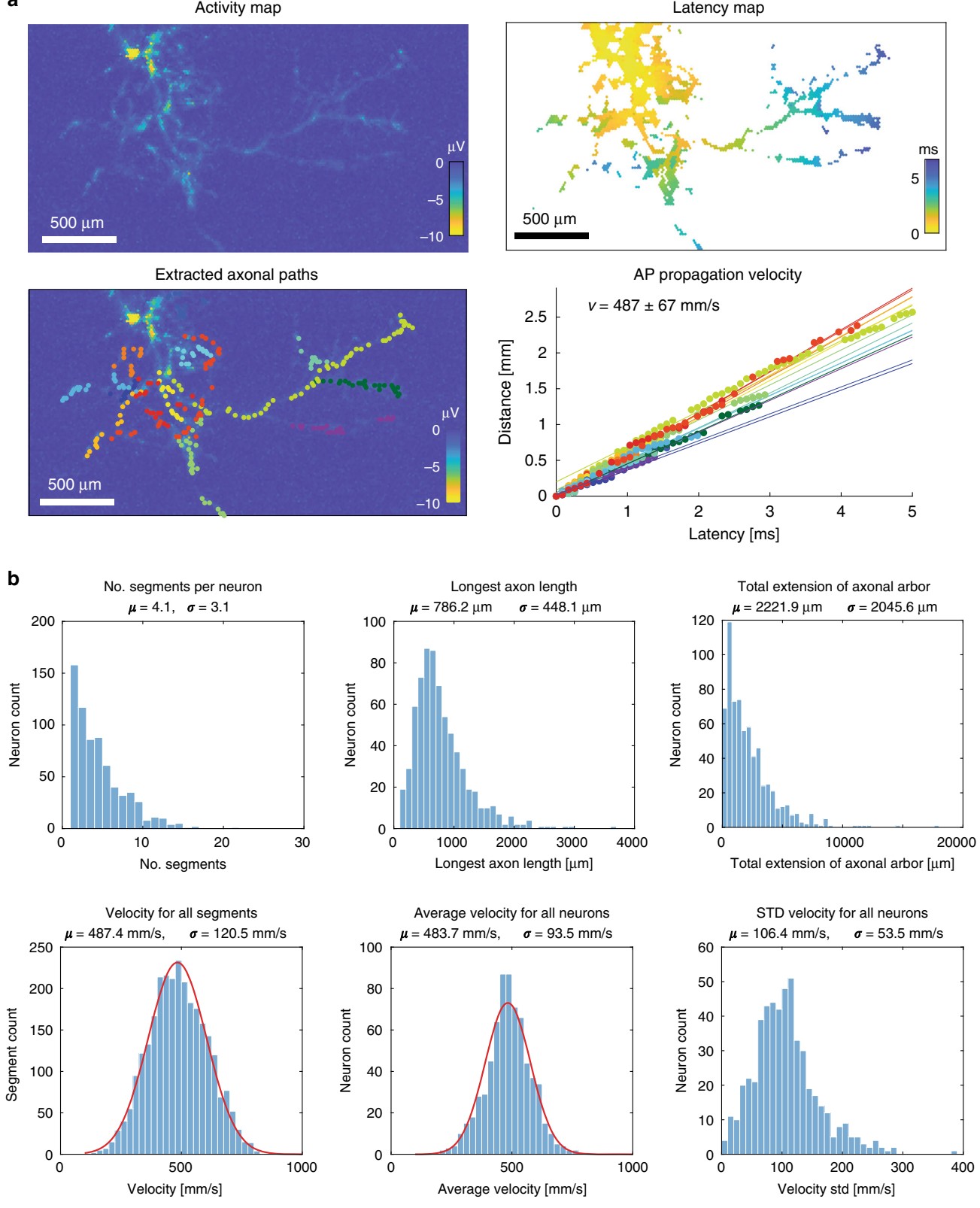

**Fig. 5 Single-cell analysis of primary-neuron cultures. a** Examples for the estimation of axonal-signal-propagation velocity of one neuron, including amplitude map, latency map, the locations of the selected segments, and a linear fit to calculate the velocity for each segment. **b** Single-neuron-derived statistics for these primary cultures (*n* = 1086 neurons, data recorded from five cultures on DIV 20), including number of axonal segments, longest axon length (longest selected segments), total extension of axonal arbors (sum of all assigned segments), and action-potential-propagation velocity for all segments, average velocity for all neurons. Mean values (*μ*) are given as well as standard deviations (*σ*). Source data are provided as a Source Data file.

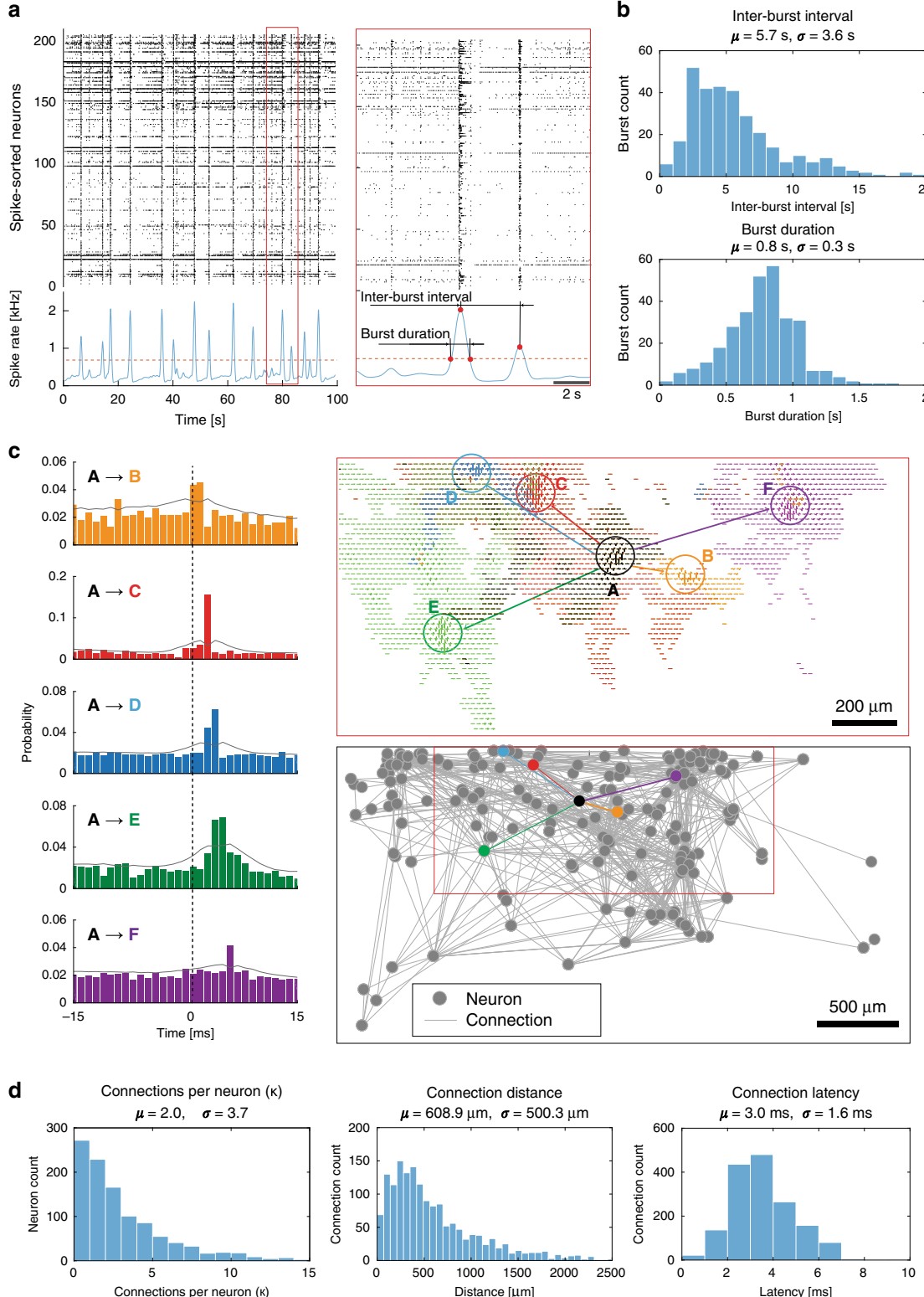

**Fig. 6 Network analysis of primary-neuron cultures. a** Raster plot of a burst ($n = 204$ neurons from one culture at DIV 20). The detection of the burst was done by thresholding at 1.35× the baseline spike rate (after convolution using a Gaussian kernel), the definition for the interburst interval and burst duration is shown in the figure. **b** Burst-related statistics have been extracted ($n = 1086$ neurons, recorded from five cultures at DIV 20), including interburst interval and burst duration. **c** Network analysis of a primary-neuron culture ($n = 204$ neurons from one culture at DIV 20), including cross-correlograms and footprints of the selected connections between the black neuron A and surrounding neurons in different colors (B-F). The network-connectivity plot shows the detected connections between all neurons in the culture. **d** Network-related statistics have been extracted ($n = 1086$ neurons, recorded from five cultures at DIV 20), including the number of connections per neuron, distance, and latency for each connection. Mean values ($\mu$) are given as well as standard deviations ($\sigma$). Source data are provided as a Source Data file.

Previous studies have demonstrated the feasibility of probing monosynaptic connections among neurons by analyzing millisecond, short-latency peaks in the pairwise cross-correlograms (CCG), inferred from their ongoing spiking activity[4]. HD-MEAs and patch clamp have been used in combination to assess and validate connectivity features[43]. We extracted the corresponding features from the large data sets. An example for a cross-correlation-based connectivity analysis of putative excitatory connections between a neuron A (in black) and five other neurons (B-F, each highlighted in a different color) is given in Fig. 6c. To test the significance of the cross-correlogram, we applied a convolution method ($\alpha = 0.01$)[44]. We considered a peak in the pairwise cross-correlogram, clearly exceeding the baseline activity and the estimated bounds, as evidence for a potential synaptic connection between the two neurons (Fig. 6c). The peak delay times, i.e., the distance of each significant peak in the cross-correlogram to 0 ms, correlated with the relative spatial distance of the neurons (Supplementary Fig. 2b; $R = 0.33$), indicating the influence of axonal signal propagation on interneuron functional connectivity. Next, we performed an all-to-all cross-correlation analysis among all detectable neurons on the DM-MEA and estimated metrics to quantify aspects of the overall network organization, including the number of synaptic connections per neuron (or degree $\kappa$), connection lengths and connection latencies (Fig. 6d). Estimated connectivity graphs were sparse (in total 410 connections for 204 neurons from one culture), and demonstrated a long-tailed degree distribution with only few highly-connected hub neurons (only 4.5% of neurons showed a degree of connectivity of more than 10; the degree of connectivity is a descriptor of the number of connections per neuron). The average number of excitatory connections per neuron was $2.0 \pm 3.7$ (DIV 20, $n = 1086$ neurons from five chips). The average connection distance between neurons was $608.9 \pm 500.3$ μm; the average peak latency was $3.0 \pm 1.6$ ms. Interestingly, we also found a correlation between the total degree and total extension of the axonal arbor ($R = 0.40$), which may indicate a potential link between neuron morphology and functional connectivity (Supplementary Fig. 2c, d).

**Characterizing neuronal network development**. Full-frame DM-MEA recordings can also be used to characterize the development of neuronal networks. For a proof-of-principle, we acquired array-wide recordings over multiple DIVs from three chips and provided summary statistics on the development of some key single-cell and network features. Figure 7a illustrates how neurons can be followed during the development at the single-cell (third column) and network level (fourth column). The presented results/metrics were pooled over 538 neurons and recorded from the three chips at DIV 10, 14, and 18 (recording duration 3 min). Full-frame activity maps, i.e., the maps showing the amplitude of APs on each electrode, indicated a clear increase in extracellular spiking activity (left panels). We inferred the spike amplitude from the largest signal amplitudes of spike-sorted neurons and increased from $90 \pm 35$ to $150 \pm 80$ μV (173%) across the nine days, so did the firing rate (increase from $1.7 \pm 2.1$ to $3.6 \pm 2.8$ Hz (212%)). Similarly, we found a strong increase in footprint extension, that is, in the average number of electrodes associated with each neuronal footprint. This increase reflects previous results from high-content imaging[31] and likely indicates the neurite outgrowth during development. The number of detected axonal segments also increased from $0.2 \pm 0.6$ to $3.6 \pm 2.7$. Moreover, we observed an increase in the total axonal arbor extension (from $0.56 \pm 0.40$ to $1.9 \pm 1.5$ mm). At the same time, the average propagation velocity increased from $380 \pm 100$ to $470 \pm 90$ mm/s (124%). A pronounced connectivity increase was observed at the network scale. The number of connections per neuron (degree) increased by about eight times from DIV 10 to DIV 18, although the change in the average length of the connections was rather moderate. Interestingly, the burst duration for all three cultures did not change significantly with development and remained around $0.8 \pm 0.2$ s.

## Discussion

We presented a highly versatile DM-MEA for live-cell analysis of neurogenic cells and demonstrated its applicability for the functional characterization of neuronal preparations at subcellular, single-cell and network levels. Ideally, HD-MEAs provide high signal-to-noise ratio for recording small neuronal signals and enable, at the same time, high-spatial-resolution measurements. In a technical realization, however, tradeoffs have to be made in designing HD-MEAs and associated circuitry. These tradeoffs are partially interdependent and concern key features of such systems: spatial resolution, temporal resolution (sampling rate), signal-to-noise characteristics, power consumption and chip real estate. Low-noise circuitry with a high sampling rate to detect small signals consumes significant power and requires considerable chip area, whereas high spatial resolution requires a dense packing of small electrodes so that only very few and small circuitry elements for addressing and signal amplification can be realized and repeated with each electrode.

The DM-MEA concept is an attempt to optimize as many features as possible considering the tradeoffs mentioned above. The full-frame APS readout features a noise level of 10.4 μV$_{rms}$, while the SM readout achieves a noise level of 3.0 μV$_{rms}$. The SM performance is comparable to current state-of-the-art devices, while the APS noise performance is significantly improved in comparison to other APS devices with a similarly high spatial resolution[19–27]. In addition, we here realized both modes in a single device, which is challenging, as both modes require dedicated circuits and consume area within each pixel. The simultaneous availability of APS and SM modes offers the possibility of detecting small signals in the SM mode and using those as trigger signals for averaging in the APS mode. Through this feature, the system is capable of recording small-amplitude signals, such as electrical signals of human iPSC-derived neurons or propagating axonal APs (<15 μV), over large active areas at high throughput, which is hardly possible using APS-only devices[20,23]. For signals with large amplitudes (>100 μV, e.g., from primary neurons), the DM-MEA can be used to assess waveform shapes using SM data with high sampling rate and high low-pass frequency, while spikes could be reliably detected and sorted from APS-mode data, which features lower sampling rate and higher low-pass frequency (Supplementary Fig. 3). We here immediately achieved the characterization of more than 1000 individual neurons (Fig. 5), along with their axonal arbors, while the number of individual neurons and processes characterized previously by using the SM alone was approximately 50[30].

The DM-MEA can sample both, population-wide LFPs and single-neuron spiking activity in the same recording, as it provides the subcellular spatial resolution and high sampling rate. In contrast to live-cell fluorescence imaging, using, for example, calcium indicators for slow network oscillations[45] or voltage-sensitive dyes for single cells[32], the proposed functional electrical-activity imaging is label-free, so that there are no phototoxic side effects, and continuous long-term monitoring of the whole sample can be achieved. Moreover, the primary electrical signals can be directly observed at sufficient spatiotemporal resolution without the need of signal transduction into another domain.

We used the DM system to record and investigate tissue-specific functional features, such as light responses of the retina or

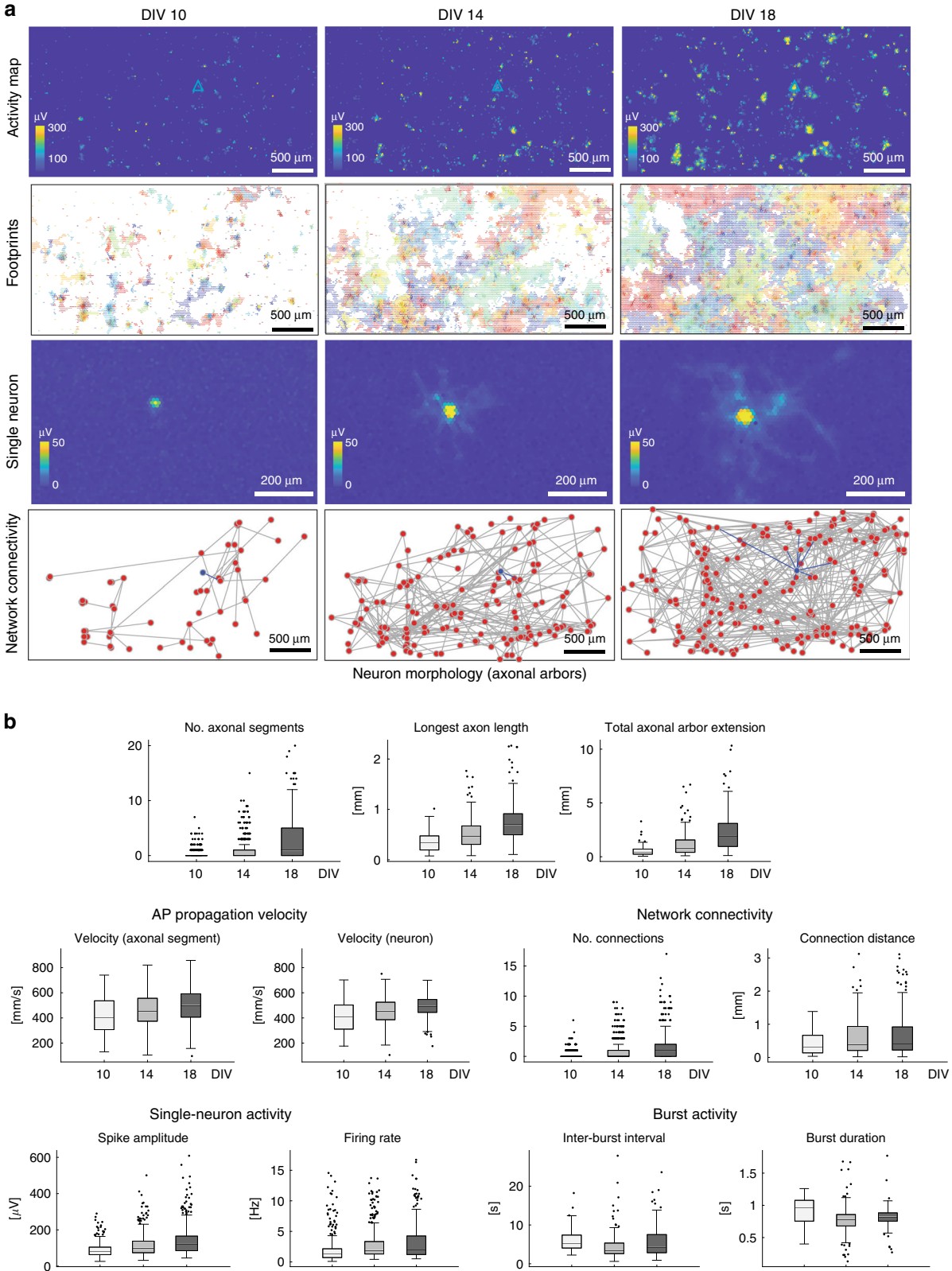

**Fig. 7 Comparison of primary rodent cultures at DIV 10, 14, and 18. a** Activity map, electrical-activity footprint map, example single-neuron footprint, and network-connectivity map of one selected culture at the different development stages (204 neurons). The location of the selected neuron has been marked in blue in the amplitude map, and the network connectivity of a selected single neuron has been indicated with blue lines in the connectivity map. **b** The statistics at the different DIVs are summarized ($n = 3$ cultures, 538 neurons), including activity-related features (spike amplitude and firing rate), single-cell related features (the number of axonal segments, total extension of axonal arbors, longest axon length and AP propagation velocity for all neurons), and network-related features (interburst interval, burst duration, number of connections per neuron, and the connection Euclidean distance). The box plots display the 25th percentile, median, and 75th percentile, and the whiskers represent ±1.5× interquartile range. Source data are provided as a Source Data file.

LFP-propagation in hippocampal slices. Functional data could be simultaneously acquired from the whole sample at high spatio-temporal resolution so that DM-MEAs enabled us to study, for example, the origin and direction of propagating network activity[33], the relationship between AP spikes and LFPs[46], and the responses of different areas of the sample to stimulation or pharmacological interventions. The SM devices can be used to study some of the mechanisms underlying specific brain states (e.g., sleep) and diseases using brain slices, cell cultures (also of human origin through the iPSC route) and other in vitro preparations.

As the DM-MEA enables features to be extracted at subcellular, single-neuron, and network level, we implemented several functional assays, including the whole-sample activity mapping, tracking of axonal-signal dynamics, or analysis of putative monosynaptic connectivity. We demonstrated simultaneous label-free tracing of the axonal arbors of hundreds of neurons in the same sample, with recordings lasting only a few minutes. Moreover, we could perform the tracking of individual neurons over multiple days (Fig. 7) by monitoring electrical activity on the array over time and localizing the corresponding electrical footprints. Compared to axonal-outgrowth results obtained by the use of imaging techniques in low-density neuron populations[47], the axonal arbors extracted from the electrical footprints here were acquired from single-cell activities so that single cells and their axonal arbors could be tracked during development. Axonal parameters, such as AP propagation velocity and extension of axonal arbors, can be used to investigate axonal dysfunction related to neurodegenerative diseases[48] and for drug testing[47], while network parameters, such as burst patterns and network connectivity, can be used to study the synaptic plasticity[4,49].

This work demonstrated large-scale extraction of axonal arbors and single-neuron activity; however, spike sorting of a multitude of single units currently faces limitations, as thousands of cells per sample need to be sorted. We expect further improvements through advancements in the development of spike sorting algorithms, in particular with respect to sorting speed and capability of automatically processing large datasets.

In summary, we presented an all-electrical approach to live-cell functional imaging and analysis with a DM-MEA that can be used to simultaneously measure subcellular, cellular and network-level features across a wide range of electrogenic cell and tissue types. Key functional electrophysiological parameters, such as axon morphology and conduction velocity, are challenging to obtain through other methods. We envision a broad applicability of the DM-MEA system to probe physiological and pathological neural activities in in-vitro systems and disease models. The DM-MEA ideally complements existing techniques, such as high content imaging and automated patch-clamp. Increasing the number of preparations that can be simultaneously recorded from, such as by arranging several DM-MEAs in a well-plate-like format, will enable researchers to perform high-throughput assays for compound testing and profiling cell lines. The platform presented here will prove useful in developing, characterizing, and validating the human iPSC models for discovering novel therapies for neurodegenerative diseases.

## Methods

**Technical details of the DM-MEA.** The DM-MEA consists of $108 \times 192$ pixels in a hexagonal arrangement[26], resulting in a total of 19,584 electrodes at an electrode pitch of 18.0 μm. The system includes 19,584 APS readout channels, sampling at 11.6 kHz, 246 SM readout channels, sampling at 24.4 kHz, and 8 stimulation buffers, each of which can drive stimulation signals for several electrodes (Fig. 1b). Every electrode can be simultaneously read out in APS and SM mode and can be used for electrical stimulation with voltage or current stimuli. The DM-MEA has been fabricated using 0.18 μm CMOS technology. The chip size is $9.0 \times 6.0$ mm$^2$ with an array size of $1.8 \times 3.5$ mm$^2$. The noise level was 10.4 μV$_{rms}$ for the APS

mode and 3.0 μV$_{rms}$ for the SM mode in the AP frequency band (0.3–5 kHz), and the total estimated power consumption was 125 mW. Labview 2017 was used for programming the device and for data acquisition. MATLAB R2019a was used for the analysis of the acquired data.

**Biological preparations.** All experimental protocols were approved by the Basel Stadt veterinary office according to Swiss federal laws on animal welfare and were carried out in accordance with the approved guidelines. Animals were kept under an inverse 12 h day-night cycle and housed in temperature-controlled ($21 \pm 2$°C) and humidity-controlled ($55 \pm 10$%) rooms. A step-by-step protocol describing the preparation of biological samples for the recording of electrophysiological signals using HD–MEAs can be found at Protocol Exchange[50].

*Primary neuronal cultures*: Primary neuronal cultures were prepared from embryonic-day-18 Wistar rat cortices using a previously published standard protocol[51]. Briefly, cortices were dissociated enzymatically in trypsin with 0.25% EDTA (Life Technologies, Bleiswijk, Netherlands) and physically by trituration. The surface of the chip was coated with 0.05% polyethylenimine (PEI, Sigma-Aldrich, USA, #181978) and laminin (Sigma-Aldrich, #L2020) to increase cell adhesion. The cells were plated on the DM-MEA at a density of 3000 cells/mm$^2$. The neuronal cultures were maintained in a humidified cell-culture incubator at 37 °C and 5% CO$_2$/95% air. All experiments with 2D cultures were conducted inside a stage-top incubator (OTOR-RE from Tokai Hit, Japan) at 37 °C.

*iPSC-derived glutamatergic neurons*: Human iPSC-derived glutamatergic neurons were purchased from Fujifilm Cellular Dynamics International (iCell GlutaNeurons Kit, USA, #01279). The neurons were prepared following the protocol provided by the supplier. The surface of the DM-MEA was coated with 0.1 mg/mL poly-D-lysine (PDL, Thermo Fisher Scientific, USA, #A3890401) and Geltrex (Thermo Fisher Scientific, #A1569601) to improve the cell adhesion. The neurons were plated onto the MEA at a density of 40,000 cells/mm$^2$.

*Brain slices*: The brain slices were prepared as previously published[52]. The brains of mice (3 weeks old) were sliced at a thickness of 200–300 μm for the cerebellum and hippocampus region. The acute slices were placed on top of the array and held in place with a membrane (Tissue Holder, MaxWell Biosystems AG, Zurich, Switzerland). During the experiments, the slices were perfused with artificial cerebrospinal fluid (in [mM]: NaCl 125, KCl 2.5, NaH$_2$PO$_4$ 1.25, MgSO$_4$ 1.9, Glucose 20, NaHCO$_3$ 25, CaCl$_2$ 2) with carbogen (95% O$_2$, 5% CO$_2$) at 37 °C with a heater (TC01, Multi-channel Systems, Reutlingen, Germany).

*Retina*: Retinae were isolated from adult Wistar rats. The eyes were dissected at room temperature under dim red light conditions in Ames solution (Sigma, A1420), which was continuously equilibrated with 5% CO$_2$ to 95% O$_2$. The vitreous was removed, and a retina patch was placed ganglion-cell-side down on the electrode array and perfused with Ames solution (pH 7.4, 35 °C), equilibrated with 5% CO$_2$ to 95% O$_2$. The retina patch was pressed onto the array with a polyester membrane (Tissue Holder, MaxWell Biosystems AG, Zurich, Switzerland). Retinal ganglion cell extracellular activity was recorded for 1–2 h. For light stimulation, a LED flashlight with light pulses with a frequency of 0.5 Hz was used.

SCAD devices (SCAD, Stem Cell & Device Laboratory, Inc., Kyoto, Japan) consist of nanofiber scaffolds that can be used to attach neuronal cells to build 3D cell cultures[39]. These scaffolds with aligned nanofibers have been prevailingly used for the creation of human-like, multilayered, and highly oriented cardiac microtissues with the aim to achieve reproducible and consistent drug responses from 3D cell cultures. The SCAD-MT$^{TM}$ cell platform has been specifically developed for efficacy, safety, and toxicology models and assays for drug discovery and development. The surface of the SCAD device was coated with 0.002% PLO (Poly-L-Ornithine, Sigma-Aldrich, P4957-50ML) and 10 μg/mL laminin (Sigma-Aldrich, L2020). IPSC-derived human dopaminergic neurons (Elixirgen Scientific, Baltimore, USA) were cultured on SCAD devices within a well-plate filled with medium using a protocol provided by the supplier. For recording, the SCAD device was placed and pressed onto the array with a membrane (Tissue Holder, MaxWell Biosystems AG, Zurich, Switzerland).

*Neuron spheroids*: The dopaminergic neuronal spheroids were prepared using a published protocol[38,53]. The human induced pluripotent stem cells (iPSCs) were induced to form dopaminergic progenitor cells (DAPs), the steps included to start with high density cultures, to perform cell sorting by using a cell surface marker for the floor plate, and to establish a maturation culture to obtain floating aggregates[53]. The surface of the array was coated with PEI and laminin to promote tissue adhesion.

**Immunohistochemistry and fluorescence imaging.** Primary neurons were fixed with 4% paraformaldehyde and immunostained with primary (antiMAP2, 1:500, Abcam, #ab5392) and secondary antibodies (goat antichicken 647, 1:200, Invitrogen, #A21449). For fluorescence imaging (Fig. 2a), we used a Nikon NiE upright confocal microscope with a Yokogawa W1 spinning-disk scan head at ×10 magnification.

**Spike sorting and spike-triggered averaging.** Due to the high spatial resolution of DM-MEAs, a single neuron is usually recorded by multiple electrodes. To clearly assign the activity to individual neurons and to determine the spatial spread of neuronal activity on the DM-MEA, we implemented a semiautomatic spike sorting.

As the use of the APS mode of the DM-MEA provides signals of 19,584 channels, which in their totality cannot be easily sorted with currently available spike sorters, we performed a preselection and included only electrodes that featured the signal amplitudes larger than 80 μV. We thus reduced the number to about 1000–1500 channels, depending on the activity level of the culture. We then spike-sorted the data sets with Spyking Circus[54], which is a semiautomatic spike sorter based on principal component analysis (PCA) and template matching. The raw data were filtered with a band-pass filter of 300–3000 Hz, and the threshold for spike detection was set to 7.5 times the standard deviation of the filtered signal. Once spikes were assigned to individual units, we performed spike-triggered averaging for all sorted units with a template length of 12 ms. Here, we used the spike times of the sorted units as trigger points to average the raw data over the entire array. Thus, we were able to increase the SNR of the averaged signals and to detect even small axonal signals. The spike-triggered extracellular signal of a sorted unit across several electrodes of the array was termed "electrical footprint".

**Inferring axonal arbors and propagation velocity.** We used spike-triggered averaging to obtain the footprint for each neuron on the array To obtain a clean footprint for estimating the extent of the axonal arbor, we separated the electrodes featuring axonal signals from electrodes recording mainly noise using a previously introduced method, which is based on the local correlation of extracellular signals[41]. The method that we used relies on the spike amplitude as well as the standard deviation of the spike times. From the extracted template, which included all electrodes, we excluded the electrodes yielding mostly noise by applying a threshold of 5.5 times the standard deviation of the signal for the spike amplitude and a threshold of 0.7 ms for the standard deviation of the peak time. The delay map in Fig. 4a shows the results for one example neuron. After the extraction of clean footprints, we estimated the extension of the axonal arbors. For this purpose, we developed and applied an algorithm that enabled automatic extraction of different axonal branches by tracking the signals moving continuously along an axon frame by frame (Fig. 4).

Two parameters were used during the extraction process: spike amplitude, which was measured at the signal peak on the respective electrode (Fig. 4a), and time delay, which represents the time difference between the signal peaks at each electrodes in reference to the signal at the initiation site (axonal initial segment, see Fig. 4c). The extraction starts at the last frame of each axonal segments to cope with signals branching into multiple segments. By starting with the last frame and tracking the signal backwards in time towards the initiation site, the number of potential axonal paths is much more limited so that they are easier to extract. For each segment, after having defined the last frame $N$ (point 1 in Fig. 4c), the extraction process started by searching for electrodes with peaks in the preceding frame $N-1$ within a spatial radius of 80 μm. Among these electrodes, the electrode with the largest amplitude was selected (point 2 in Fig. 4c). Then, the same procedure was applied to the next frame $N-2$ and was continued until reaching electrodes close to the signal initiation site. The procedure was stopped five frames before the initiation site to avoid the inclusion of signals of the soma or the AIS. The signals generated by the AIS are large and visible on many electrodes, so that precise signal timings are difficult to assess in that area. Since axonal signal generally feature low amplitudes of 5–10 μV, detection may become difficult, and gaps may occur during the tracking procedure (e.g., in Fig. 4c from 20 to 21). In this case, the search of preceding signal was expanded to $N-R$ frames (up to $R = 5$), and the searching radius was expanded to $R \times 80$ μm. Thus, it was possible to continue the axonal tracing and to obtain axonal paths without interference of signals from axonal branches of nearby or neighboring neurons. The extraction procedure stopped upon reaching the time point of five frames before the signal initiation, or if no more suitable electrodes could be found.

For each detected axonal segment, the distances between the identified electrodes were summed to obtain the total distance that the action potential had traveled from the initiation site. To calculate the velocity in each segment, we performed a linear fit of the interelectrode distance versus the spike-time latency (Fig. 4e).

**Inferring monosynaptic connectivity and burst statistics.** Based on the spike trains of all spike-sorted neurons, we used a baseline-corrected cross-correlation method that has been established to infer monosynaptic connectivity in networks[4]. We calculated the cross-correlation (1 ms binning) of spike trains between every two neurons, and the resulting CCG was convolved with a Gaussian kernel as bounds[44]. The synaptic connections were detected, when the peaks in the CCG exceeded the convolution bonds within ±5 ms, and the directions were defined by the polarity of the detected peaks (negative peaks represent presynaptic connections, and positive peaks represent postsynaptic connections). The connection latency was defined by the position of the peaks in CCG, and the connection distance was defined by the distance between the centers of the two connected neurons.

Bursts were detected on spike-sorted neurons within the same culture, while five cultures were measured to extract burst-related statistics in Fig. 6b. We calculated the synchronized firing rate by performing, first, an integration of individual spikes from all neurons with a binning of 0.01 s, and, second, a convolution using a Gaussian kernel (standard deviation = 0.3 s). A burst was detected as synchronized co-activity of neurons over a threshold of 1.35× the baseline spike rate, and the peaks of the burst were detected for the calculation of burst duration. The definition of the interburst interval and burst duration is illustrated in Fig. 6a.

## Data availability
The data that support the findings of this study are available in the ETH Research Collection (https://doi.org/10.3929/ethz-b-000431730). Source data are provided with this paper.

## Code availability
The developed code for the axonal-arbor tracing is available at: https://github.com/xyuan-github/axon.

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

## Acknowledgements

We thank Silvia Ronchi, Dr. Vishalini Emmenegger and Annalisa Bucci, all at ETH Zurich, for technical support with sample preparations, and the BSSE cleanroom staff Peter Rimpf and Tomislav Rebac for post-processing of CMOS chips. Financial support through the European Research Council Advanced Grant 694829 "neuroXscales" and the corresponding proof-of-concept Grant 875609 "HD-Neu-Screen", through the EU Marie Skłodowska-Curie Individual Fellowship Grant 798836 "MAPSYNE", through the Swiss Innosuisse Project CTI No. 25933.2 PFLS-LS and through the Swiss National Science Foundation under contract 205320_188910/1 is acknowledged.

## Author contributions

X.Y. designed and conducted experiments, analyzed the data; X.Y. and U.F. developed CMOS circuitry and the data acquisition system; M.S. provided support for data analysis; M.E.O., M.F., W.G., T.K., A.O., S.N., I.S., and J.T. provided experimental support; X.Y., M.S., M.E.O., U.F., and A.H. wrote the manuscript; A.H. and U.F. supervised the project.

## Competing interests

The authors declare the following competing interests: M.O., M.F., and U.F. are co-founders of MaxWell Biosystems AG, which commercializes HD–MEA technology. The remaining authors declare no competing interest.
