## [Peer Review File · Nature Communications]

Reviewers' Comments:

Reviewer #1:

Remarks to the Author:

Reviewers Comments: NCOMMS-20-16492-T, Yuan et al., "Versatile live-cell activity analysis platform for characterization of neuronal dynamics at single-cell and network level"

These authors report a variety of data from a dual mode multielectrode array DM MEA, that can operate in a fast scan mode of the ~20,000 sites at 11kHz, or a mode with 246 electrodes sampled at ~24 kHz simultaneously. They do not describe the circuitry in any detail. The device seems to be reported in IEEE Biomed Circuits Syst Conf. . 2019 Jun 18; 2018:BIOCAS.2018.8584735, and very oddly the authors do not cite that report. Aside from a few awkward English constructions the paper is clearly written and well-illustrated. While these constructs are a bit jarring to a native English speaker, they do not interfere with technical understanding. I note some specific issue below.

I have two concerns with this manuscript. Much of the capability reported here has been previously published by one or another of the authors. This is especially true of neuronal process imaging and spike time averaging. As far as I can tell, the entirely new information is the application to iPSC's. The most impressive use for this sample in the assignment of the 246 switched mode channels so that the APS mode can signal average across the whole array and spike time averages extracted from the relevant channels. However, the authors do not claim the simultaneous operation of the APS and DM operations is needed to accomplish this data. It may be that this is unique data but my literacy on the use of neuronal iPSC's is not deep enough to know if any other reports have been made. I do not find any in the principal authors record.

Second, the authors go to great length to illustrate the neuronal morphology functionality of the data while ignoring the brilliant images that could be made with widely available microscopy. A simple DIC image with a water immersion objective could capture a substantial fraction of the device in a second with far greater morphology detail than the 18 um pitch electrodes. The brilliant image in Figure 1a, makes this argument far better than I can. Even for the SW scan to fund the 246 electrodes for spike detection in iPSC, a simple image would find all the cells quickly with much less time and ambiguity.

Finally, I would like to ask the authors to include cell density for each of their illustrations. Many look like extremely sparse cell seeding. If most of these applications require that only a handful of cells be present, is the device valuable for the applications proposed?

There is likely useful material here, but the applications list reads more like a brochure for Maxwell Biosystems than a scientific report of novel results. If this dual scan device can perform previously reported measurements as well as prior reports, a single sentence would suffice.

Reviewer #2:

Remarks to the Author:

Authors report the application of dual-mode(DM) high-density CMOS MEA chip for the analysis of

neural preparations in vitro. The DM chip has been developed to combine both APS and SM type CMOS MEA chips, which have their own pros and cons. According to the supplementary data (manuscript), the specification (noise level, frame rate, etc.) is very impressive.

In this reviewer's point of view, this manuscript shows a new possibility of CMOS MEA chip in neuroscience studies. It allowed large-area, high-resolution, multi-scale (single cell and network level) analysis of cultured neuronal networks. According to the demonstrations, it would be also possible to produce similar quality outcome for brain slice preparations.

There are some major issues that need to be addressed before publication:

1. In introduction part, iPSCs were focused too much, although there were only a small demonstration with iPSCs. Please rewrite the introduction part to be more relevant with main results.
2. Were all data presented (figure 3 ~ 7) used dual mode concept shown in Figure 2? How critical was it to combine APS and SM mode in obtaining the data? Can you show the difference (DM vs APS vs SM)?
3. Figure 1 should contain more information about the DM-MEA. Some information (schematics, functional block diagram, ...) that should be needed to understand the key concept. Remove part (b) which need references for each prep. Not sure if this will help this manuscript. It may be more suitable for a review article.
4. Figure 2 (key concept) seems to be important to understand how authors used dual mode to acquire the data. Please add similar experiment timeline for other figures (figure 4, 5, 6, 7). Were figure 4,5,6,7 all done with dual mode? Or single mode? No information could be found.
5. Figure 3: Having many showcases distracts the main originality of this work. Each showcase only serves as a demonstration and it is difficult to appreciate the quality of the data. Each of them has been shown by single mode chips (authors group and others). I don't believe that DM chip only allows all preparations.
6. Figure 4 & 5: Figure 5 should come first with Figure 4a, so that the actual analysis technique can be highlighted. Then, figure 4b (results).
7. The following claim requires more data analysis (in discussion): "... demonstrated its applicability for the functional characterization of a broad range of biological preparations. ..." —> only cultured neurons were analyzed in detail. Others were just showing that it 'may' be possible. It was merely a 'showcase' as authors mentioned in main text.
8. There is no biological validation data for assays. Can authors point out their previous works that validate the assay design using CMOS MEA chips? For example, does the axonal arbor coincide with actual axonal growth?
9. Figure 7: I think is the most striking data set from this work. Authors should focus more on the capability of multi-scale network analysis: single cell axonal branching and network. Figure 7 is a very unique data set that would be very difficult to obtain by conventional experimental platforms. Figure 7b should be rearranged such that single cell structure and function, network structure and function are well high-lighted.

Reviewer #3:

Remarks to the Author:

The paper by Yuan et al describes the use of a dual-mode, dense microelectrode in vitro recording system. Multiple types of tissue preparation are used and some fairly sophisticated analyses are applied to demonstrate the utility of the device and potential for new lines of neurophysiological investigation. The work appears to be rigorous and the paper is well-written. My comments are all fairly minor and aimed mostly at improving the presentation.

Introduction- The emphasis here is only on iPSC. This is shortchanging the more extensive work presented in the paper on organotypical slices, primary cell cultures, and even a 3D culture substrate. All of these preparations have their own advantages and should be mentioned in the introduction with a little less emphasis on iPSC.

A disadvantage of imaging studies is their temporal imprecision. This might be mentioned.

An 11 kHz sampling rate is very low for discriminating and sorting extracellular action potential waveforms (unless those in culture are much wider than those in vivo). This should probably be addressed, as well as the high-pass cutoff of 3 kHz which also seems low.

In the initial description of the dual-mode capability, it is important to stress that each electrode can be recorded using both modes simultaneously. This is mentioned in the Methods but not until then.

The heading on line 276- "Mapping dynamics..." – bursting characteristics are not really considered to be a dynamical feature for many neurophysiologists. Different wording would be helpful.

Wording issues (minor)

The constructs "enable to" and "allow to" are used throughout the document and this seems incorrect. For instance:

line 60- "enable to simultaneously record from" could be rewritten as "enable simultaneous recording from"

line 71- "allows to pack electrodes very close to each other" to "permits dense electrode packing"

line 379 "the DM-MEA enables to extract features", could be changed to "the DM-MEA enables features to be extracted"

The sentence in lines 66-70 is rather long. Suggestion: "... addressing scheme within the array: the front-end amplifiers..."

Reviewer #1

These authors report a variety of data from a dual mode multielectrode array DM MEA, that can operate in a fast scan mode of the ~20,000 sites at 11kHz, or a mode with 246 electrodes sampled at ~24 kHz simultaneously.

We thank the reviewer for the comments on the work.

They do not describe the circuitry in any detail. The device seems to be reported in IEEE Biomed Circuits Syst Conf. . 2019 Jun 18; 2018:BIOCAS.2018.8584735, and very oddly the authors do not cite that report.

We thank the reviewer for pointing out this omission. We apologize for missing this point and added the citation to the manuscript (now reference 28). The report in IEEE BioCAS is a brief conference contribution. We are currently working on a manuscript explaining all circuits and the electronic-system design in detail, which we will submit to a specialized circuitry journal within the next weeks. We have now added a schematic that explains the DM concept as new Figure 1b.

Aside from a few awkward English constructions the paper is clearly written and well-illustrated. While these constructs are a bit jarring to a native English speaker, they do not interfere with technical understanding. I note some specific issue below.

We have improved the English writing after proof-reading by a native speaker. Changes include, amongst others,

- Less use of passive voice
- Correct usage of comma, colon and semicolon
- Rephrasing of several sentences

I have two concerns with this manuscript. Much of the capability reported here has been previously published by one or another of the authors. This is especially true of neuronal process imaging and spike time averaging. As far as I can tell, the entirely new information is the application to iPSC's. The most impressive use for this sample in the assignment of the 246 switched mode channels so that the APS mode can signal average across the whole array and spike time averages extracted from the relevant channels. However, the authors do not claim the simultaneous operation of the APS and DM operations is needed to accomplish this data. It may be that this is unique data but my literacy on the use of neuronal iPSC's is not deep enough to know if any other reports have been made. I do not find any in the principal authors record.

As recent advances in iPSC technology provide a promising approach to study neurological disorders in vitro, we expect that there will be an increasing demand for comprehensive characterization and high-throughput functional screening of iPSCs across different scales (subcellular-compartment, individual-neuron, and network characteristics).

Low-density MEAs (with 64 or less electrodes) have been used for functional characterization of iPSC-derived neuronal cultures with a focus on spiking activities¹ (e.g. firing rate, inter-spike interval coefficient of variation) and network activity² (e.g., burst frequency, burst patterns). These MEAs, however, do not provide a lot of details on single-cell and/or network physiology, let alone subcellular-compartment features, such as axonal propagation velocity.

A major difficulty for studying iPSC-derived neurons is the apparent small amplitude of the extracellular signals ($\sim 30 \mu\text{V}$)², which can only be detected using readout schemes featuring a high SNR, i.e., RMS-noise levels of less than $10 \mu\text{V}$, which is not the case for any of the published full-frame HD-MEA systems³⁻⁶. With the dual-mode (DM) system, the experimenter can use the high SNR signals, recorded in switch-matrix (SM) mode simultaneously in the same culture, to extract spike times and trigger points to average the signals captured in APS full-frame mode. This way, a comprehensive, quick and efficient electrophysiological characterization of the entire culture can be achieved with DM-MEAs. A simultaneous operation of APS and DM modes in the same preparation

is needed and pivotal, and it is the first time that we recorded such data sets combining both modes. We have modified the manuscript to make these points more clear. (line 110-118, line 396-401)

We agree with the reviewer that neuronal-process imaging has been previously reported by our group using other devices⁷⁻¹⁰. We would like to emphasize, however, that the availability of the dual-mode (DM) system drastically improves the efficiency for these analyses, as it is no longer necessary to scan through a number of electrode subsets sequential (e.g. 26 recordings (with 1024 electrodes each) to record the full 26'400-electrode MEA introduced in ref. ¹¹).

The number of individual neurons and processes that have been characterized and reported in earlier publications⁹ was approximately 50, while we here immediately achieved the characterization of several hundreds to thousands of individual neurons, along with their axonal arbors, at once. The combined use of SM and APS or dual-mode operation drastically improved efficiency and enabled the characterization of large numbers of neurons. We clarified and emphasized this point now in the Discussion. (line 405-408)

While the proof-of-concept studies with few neurons published earlier are certainly very important milestones, we consider scaling up to larger numbers of electrodes, more neurons, and to a broader set of metrics, as a very important step forward in order to obtain reliable datasets and statistical power required for pharmacological and/or toxicological studies.

Second, the authors go to great length to illustrate the neuronal morphology functionality of the data while ignoring the brilliant images that could be made with widely available microscopy. A simple DIC image with a water immersion objective could capture a substantial fraction of the device in a second with far greater morphology detail than the 18 um pitch electrodes. The brilliant image in Figure 1a, makes this argument far better than I can. Even for the SW scan to fund the 246 electrodes for spike detection in iPSC, a simple image would find all the cells quickly with much less time and ambiguity.

We agree with the reviewer that imaging techniques have been widely used to capture the morphology of the neurons at high spatial resolution. However, current optical imaging techniques are mainly used to assess neurite growth, morphology and cell viability, without providing information about functional metrics of the cells. Morphological imaging alone may also not be sufficient to characterize activities of different axons in bundles or changes thereof upon application of chemicals or compounds. This holds particularly true for iPSC neuronal cultures that have been shown to form cell clusters later during development, thus severely complicating the extraction of morphological features of neurons using optical means. Using calcium indicators and genetically encoded voltage indicators (GEVI), neuronal activity can be imaged^{12,13}, yet with limitations in the temporal resolution. Imaging techniques mostly require immunostaining to ensure high spatial resolution, which poses problems for tracking cells in continuous and long-term studies as a consequence of phototoxic effects arising from the fluorescence labels^{14,15}. Usually cells can be imaged for a few minutes before letting them recover for several hours in the dark.

Using HD-MEAs, we were able to measure electrical activity and extract functional metrics from neuronal preparations, such as AP propagation velocity, transmission delays between cells or network connectivity, or changes in these metrics upon drug dosage, all of which cannot be obtained with traditional optical imaging. Of course, as shown in Figure 2a, optical images can be and are used to help assessing morphological features of neurons and networks. Using “electrical imaging” by means of HD-MEAs, we were able to track the axonal arbors of individual neurons over a large area and despite the occurrence of axon bundles in neuronal cultures (Figs. 4-5), which is difficult to achieve with imaging techniques alone, as axons feature average diameters around 200 nm. Moreover, we were able to continuously record from the same neuronal preparation and extract functional and morphological metrics over multiple days in the incubator (Fig. 7). Therefore, we are convinced that dual-mode HD-MEA technology is a useful contribution and complement to optical imaging and extends the range of available options for neuroscientists.

We better clarified this point at several instances in the manuscript by making clear that HD-MEAs provide functional-imaging capabilities. (line 68-71, line 57-60, line 416-420)

Finally, I would like to ask the authors to include cell density for each of their illustrations. Many look like extremely sparse cell seeding. If most of these applications require that only a handful of cells be present, is the device valuable for the applications proposed?

We have added the cell-density for every culture preparation. For primary cultures, we used a cell seeding density of 3000 cells/mm², with a total of 20k cells being seeded in each HD-MEA well. For iPSC-derived neuronal cultures, we used a cell seeding density of 40,000 cells/mm², with a total of 260k cells being seeded in each HD-MEA well. We have added this information in the manuscript to make this point clear (line 202, line 498, line 507).

There is likely useful material here, but the applications list reads more like a brochure for MaxWell Biosystems than a scientific report of novel results. If this dual scan device can perform previously reported measurements as well as prior reports, a single sentence would suffice.

To avoid misunderstandings, we would like to clarify that the DM device is not commercially available and does not form part of the MaxWell Biosystems product portfolio. We have gone through the manuscript and modified it in several instances to tune down or remove potential brochure-like phrases and to convey a realistic estimation of the system features, its versatility and potential applications to the reader. A very powerful feature of the DM device is that a comprehensive characterization across different levels can be efficiently achieved in the same neuronal culture. Examples include a high enough SNR to detect iPSC-neuron spikes along with high enough spatial resolution to accurately trace axonal spikes or the use with retinal applications, where the APS mode allows for very fast, large-area screening of the entire retinal piece and the SM mode enables high-performance spike-sorting of selected cells.

To clarify this point, we added a section in the Discussion that explains the many trade-offs in the design of HD-MEAs, including spatial resolution, SNR, power consumption and area of such systems (line 373-383). The DM-MEA concept is an attempt to optimize as many features as possible. The performance of both, the APS and SM readout circuitry of the DM-MEA are comparable to those of state-of-the-art single-mode devices^{3-6,16-20}, or better in comparison to other APS devices³⁻⁶, however, the new combination enables a broader range of applications (Fig. 3).

Reviewer #2

Authors report the application of dual-mode (DM) high-density CMOS MEA chip for the analysis of neural preparations in vitro. The DM chip has been developed to combine both APS and SM type CMOS MEA chips, which have their own pros and cons. According to the supplementary data (manuscript), the specification (noise level, frame rate, etc.) is very impressive.

In this reviewer's point of view, this manuscript show a new possibility of CMOS MEA chip in neuroscience studies. It allowed large-area, high-resolution, multi-scale (single cell and network level) analysis of cultured neuronal networks. According to the demonstrations, it would be also possible to produce similar quality outcome for brain slice preparations.

We thank the reviewer for her/his positive feedback on our work.

There are some major issues that need to be addressed before publication:

1. In introduction part, iPSCs were focused too much, although there were only a small demonstration with iPSCs. Please rewrite the introduction part to be more relevant with main results.

We expect the DM-MEA to be particularly useful for applications involving human iPSC-derived neurons because of the combination of low-noise / high-SNR readouts, as achieved by the SM readout, and the full-frame readout made possible by the APS readout. Following the suggestion of the reviewer, however, we have modified the Introduction to introduce the other applications of the system more clearly. This includes use of primary neuronal

cultures and other in-vitro preparations, and to better reflect the contents of the paper focusing on multi-scale functional characterization of neuronal cultures. (line 30-34, line 41-44, line 57-60)

2. Were all data presented (figure 3 ~ 7) used dual mode concept shown in Figure 2? How critical was it to combine APS and SM mode in obtaining the data? Can you show the difference (DM vs APS vs SM)?

Developing human iPSC-derived neurons are an ideal test case to demonstrate the features and advantages of DM-MEAs (i.e. of using APS and SM modes simultaneously) since these cells show considerably small AP amplitudes due to their still immature developmental status. As shown in our manuscript (Fig. 2), despite their small signal amplitude, it was possible to detect the electrical activity of individual iPSC-derived neurons in DM and to reconstruct features, such as axon morphology using appropriate analyses methods. Such analyses would not have been possible with previously published APS-only MEAs.

Principally, the DM method could be applied to all other cells and/or tissue samples used in the study, however, as many preparations featured larger signal amplitudes ($>100 \mu\text{V}$), they could readily be recorded in APS mode. We would like to point out that the APS circuitry introduced in this study features significantly lower noise levels compared to previous APS devices³⁻⁶. For this reason, we used data obtained from the APS mode for most of the subsequent analyses displayed in Fig. 3-7. However, we expect the DM mode to substantially improve the characterizations of iPSC-derived neurons and any other preparation featuring small signal amplitudes.

Following the reviewer's suggestion to clarify on the use of DM, we added a new figure (Suppl. Figure 3, see below). This figure shows the difference in using the different modes (DM vs. APS vs. SM) for the extraction of electrical footprints using spike-triggered averaging. The data was acquired using DM, with simultaneous recordings in APS and SM mode. Signals of the same neuron were extracted using the three different modes from a simultaneous 20-second recording in APS and SM modes.

* The noise level for SM mode with averaging 5 APs is $1.4 \mu\text{V}$, which is identical to that of the APS mode averaging 50 APs. These numbers and noise values were used for the calculation of recording time at the bottom of the table.

** Recording time required to collect footprints from 100 neurons with an assumed minimum firing rate of 3 Hz.

*** For the SM mode, we assumed that 100 channels were used to record from 100 defined neurons, while the other channels were used to scan through the array applying 134 configurations of 1.67 s duration per configuration to record 5 spikes for spike-triggered averaging to achieve a noise level of $1.4 \mu\text{V}$.

**** For the APS mode, one full-frame recording of 16.7 s duration was necessary to record 50 spikes for spike-triggered averaging to achieve the same noise level of $1.4 \mu\text{V}$.

Suppl. Figure 3. Comparison of signal averaging results using the different modes (DM, APS and SM). (a) Electrical footprints of one neuron acquired using the DM mode. (b) Averaged waveforms of three selected electrodes (el 1-3, colored background in panel a) acquired using the three different modes (DM, APS and SM). DM: averaging of APS data with triggers from SM data (67 spikes); APS: averaging of APS data with triggers from APS data (66 spikes); SM: averaging of SM data with triggers from SM data (67 spikes). The waveforms are almost identical, while there are differences in noise levels and signal-to-noise (SNR) characteristics ($1.34 \mu\text{V}$ for DM with 67 averaged APs, $1.36 \mu\text{V}$ for APS with 66 averaged APs, and $0.44 \mu\text{V}$ for SM with 67 averaged APs). (c) Effect of averaging for the three selected electrodes, showing both, raw data and averaged waveforms with different numbers of averaged APs (1, 5, 50). (d) Summary of noise levels and SNR for SM and APS mode with different numbers of averaged APs. The noise levels were measured and averaged over multiple active electrodes (100 electrodes for APS, 50 electrodes for SM). Since DM and APS both use APS data for averaging, the SNR values with different number of averaged APs for the DM mode are identical to the APS mode. In addition, we also estimated the recording time needed to collect footprints from 100 neurons using APS and SM modes.

For the DM mode, the spikes were detected in the SM data, which then were used for averaging of the APS data; we could detect 67 spikes in the SM data. For the APS mode, the spikes were detected in the APS data and used for averaging of APS data; we could detect 66 spikes in the APS data. For the SM mode, the spikes were detected in the SM-data and used for averaging of SM data. We compared the averaged waveforms of four selected electrodes (Suppl. Fig. 3b, el 1-3, colored background in Suppl. Fig. 3a), which featured nearly identical shapes with slightly different noise levels ($1.34 \mu\text{V}$ for DM with 67 averages, $1.36 \mu\text{V}$ for APS with 66 averages, and $0.44 \mu\text{V}$ for SM with 67 averages). Based on these results, we expect that all the methods that we developed (Fig. 3-7) will work well with both DM and APS data.

In addition, we compared the signal-to-noise ratio (SNR) performance of the different modes by averaging increasing numbers of APs, i.e., 1, 5 and 50, obtained using SM and APS (Suppl. Fig. 3c-d). For this calculation, since DM and APS both use APS data for averaging, the SNR for DM mode is identical to that of the APS mode. We also estimated the recording time needed to collect footprints of 100 neurons, which highlighted the advantage of using APS mode in comparison to the SM mode.

3. Figure 1 should contain more information about the DM-MEA. Some information (schematics, functional block diagram, ...) that should be needed to understand the key concept. Remove part (b) which need references for each prep. Not sure if this will help this manuscript. It may be more suitable for a review article.

As suggested by the reviewer, we added an overview schematic in Figure 1 (Fig. 1b, see below) and modified the corresponding text in the paper to demonstrate and explain the key concept of the DM-MEA. Previous Figure 1b was shifted to the Results section as a summary graph in Figure 3 to demonstrate the wide applicability of the system.

Figure 1. Dual-mode microelectrode array (DM-MEA) system, applications and readouts. (a) Overview of the HD-MEA system, including the packaged device used for biological experiments, a micrograph of the CMOS DM-MEA and a scanning-electron micrograph (SEM) of primary neurons cultured on top of the DM-MEA. (b) Schematic concept of the dual-mode readout. Every electrode is continuously read out in the APS mode (red), while a few selected electrodes are simultaneously read out in the switch-matrix mode (purple). (c) Experimental work flow using the DM-MEA, including acquisition of electrophysiological signals with different features, and data analysis to extract various features.

4. Figure 2 (key concept) seems to be important to understand how authors used dual mode to acquire the data. Please add similar experiment timeline for other figures (figure 4, 5, 6, 7). Were figure 4,5,6,7 all done with dual mode? Or single mode? No information could be found.

The results depicted in Figures 3-7 are derived from primary neuronal cultures with large AP amplitudes. With these analyses, we would like to showcase the capabilities of the chip (inferring neuronal morphology, network analysis, etc.). Here it was sufficient to record in APS mode (as demonstrated with Fig. 2b and Supplementary Figure 3). The timeline of the other experiments is the same as in Figure 2b. We have added this information in the manuscript to make this point clear (line 269-271).

5. Figure 3: Having many showcases distracts the main originality of this work. Each showcase only serves as a demonstration and it is difficult to appreciate the quality of the data. Each of them has been shown by single mode chips (authors group and others). I don't believe that DM chip only allows all preparations.

Figure 3 was intended to demonstrate the applicability of the system, and especially its capability to record from large intact tissue samples (retina, hippocampal slice). We agree with the reviewer that some of the displayed applications could be performed with single-mode devices, however, we think that it is very advantageous to be able to assess different features across several scales in the same neuronal culture and within the recording session. Besides iPSCs, examples may include applications in the retina, with the APS allowing for very fast large-area coverage of the retinal tissue and the SM mode enabling high-performance spike-sorting of selected cells. Efficiently extracting axonal morphology and velocity from hundreds of RCGs in the retina has not been shown previously to the best of our knowledge. Similar considerations hold for a hippocampal slice, where the layer of dead cells between electrodes and preparation, which requires high S/N recording to identify single units and axons. One could use APS mode to record waves propagating across the entire slice corresponding to population activity, and correlate these waves to unit-activity identified with DM mode. Finally, the SCAD example shows, to the best of our knowledge, for the first time, that axonal signals can be tracked with HD-MEAs in single trials (without averaging, Suppl. Movie 4).

Moreover, the noise level of the APS mode of the device is considerably lower than that of other APS devices featuring similar spatial resolution (approx. 50% of the RMS noise reported in refs.³⁻⁶), which enables large-scale in-depth analyses. We have moved Figure 1b (of the first version of the manuscript) to Figure 3 (of the current manuscript) to demonstrate the scales of the different preparations with different signal properties. Moreover, we have included references for each preparation.

Finally, it is also important to note, that the design of HD-MEA systems requires various tradeoffs and not all specifications, such as low noise, low power, high spatial resolution, etc. can be optimized at the same time. With the DM-device, we aimed at optimizing key specifications to cover an as broad as possible application range. We agree with the reviewer, that some of these applications have been previously shown with other devices, but we also think that there may be value in “redoing” experiments within this wide range, as new and comprehensive information can be gained with the new system. We included a few sentences in the manuscript to highlight important key-specification tradeoffs in the circuit design of the system in the Discussion (line 373-383, line 392-394) and more clearly worked out the novel features of the system concerning various preparations. (line 208-213, line 226-228, line 237-242, line 248-249, line 262-264)

6. Figure 4 & 5: Figure 5 should come first with Figure 4a, so that the actual analysis technique can be highlighted. Then, figure 4b (results).

We agree with the reviewer that the analysis techniques should come before the results, and changed the order of Figure 4 and Figure 5 accordingly.

7. The following claim requires more data analysis (in discussion): “ ... demonstrated its applicability for the functional characterization of a broad range of biological preparations. ...” —> only cultured neurons were analyzed in detail. Others were just showing that it ‘may’ be possible. It was merely a ‘showcase’ as authors mentioned in main text.

We agree with the reviewer that only cultured neurons were analyzed in detail (Figs. 4-7). We have recorded electrical activity from various preparations (Fig. 3). We applied, for example, the axonal tracking method also for pieces of the retina as shown in Supplementary Figure 1, which provides evidence for functional characterization of other preparations. Nevertheless, we tuned down the sentence indicated by the reviewer. It now reads as follows:

“...demonstrated its applicability for the functional characterization of neuronal preparations at subcellular, single-cell and network levels.” (line 371-373)

8. There is no biological validation data for assays. Can authors point out their previous works that validate the assay design using CMOS MEA chips? For example, does the axonal arbor coincide with actual axonal growth?

We agree with Reviewer 2 that relevant references for the validation of the presented analyses/assays should be provided. We have added details on the previous publications and biological findings in the manuscript.

The assays presented in the work have been validated before using CMOS HD-MEAs:

1) the tracking of axonal arbors has been validated using high-density MEAs together with imaging to prove the position of the axons^{7,10,21}. (line 275-276)

2) the network features have been extracted using HD-MEAs together with patch clamp to validate synaptic network connectivity²². (line 318-320)

While in previous work^{7,10}, we were able to extract only a few cells and compare the extracted arbors to microscopy images in a manual fashion, the throughput that we achieved now with the DM device, will allow for validating, e.g., axonal-arbor metrics on a larger scale.

9. Figure 7: I think is the most striking data set from this work. Authors should focus more on the capability of multi-scale network analysis: single cell axonal branching and network. Figure 7 is a very unique data set that would be very difficult to obtain by conventional experimental platforms. Figure 7b should be rearranged such that single cell structure and function, network structure and function are well high-lighted.

We thank the reviewer for the positive feedback. According to the suggestion, we have modified and re-arranged the panels in Figure 7b to show and group the features at the different levels (see below). We have also modified the text in the Discussion to highlight the multi-scale analysis. (line 434-437, line 441-444)

Figure 7. Comparison of primary rodent cultures at DIV 10, 14 and 18. (a) Activity map, electrical-activity footprint map, example single-neuron footprint, and network-connectivity map of one selected culture at the different development stages (204 neurons). The location of the selected neuron has been marked in blue in the amplitude map, and the network connectivity of a selected single neuron has been indicated with blue lines in the connectivity map. (b) The statistics at the different DIVs are summarized ($n = 3$ cultures, 538 neurons), including activity-related features (spike amplitude and firing rate), single-cell related features (the number of axonal segments, total extension of axonal arbors, longest axon length and AP propagation velocity for all neurons), and network-related features (inter-burst interval, burst duration, number of connections per neuron and the connection Euclidean distance).

Reviewer #3

The paper by Yuan et al describes the use of a dual-mode, dense microelectrode in vitro recording system. Multiple types of tissue preparation are used and some fairly sophisticated analyses are applied to demonstrate the utility of the device and potential for new lines of neurophysiological investigation. The work appears to be rigorous and the paper is well-written. My comments are all fairly minor and aimed mostly at improving the presentation.

We thank the reviewer for her/his positive feedback on our work.

Introduction- The emphasis here is only on iPSC. This is shortchanging the more extensive work presented in the paper on organotypical slices, primary cell cultures, and even a 3D culture substrate. All of these preparations have their own advantages and should be mentioned in the introduction with a little less emphasis on iPSC.

We expect the DM-MEA to be particularly useful for applications involving human iPSC-derived neurons because of the combination of low-noise / high-SNR readouts, as achieved by the SM readout, and the full-frame readout made possible by the APS readout. Following the suggestion of the reviewer, however, we have modified the Introduction to introduce the other applications of the system more clearly. This includes use of primary neuronal cultures and other in-vitro preparations, and to better reflect the contents of the paper focusing on multi-scale functional characterization of neuronal cultures. (line 30-34, line 41-44, line 57-60)

A disadvantage of imaging studies is their temporal imprecision. This might be mentioned.

We have added this information to the manuscript as suggested. (line 53-56)

An 11 kHz sampling rate is very low for discriminating and sorting extracellular action potential waveforms (unless those in culture are much wider than those in vivo). This should probably be addressed, as well as the high-pass cutoff of 3 kHz which also seems low.

We agree with the reviewer that a higher sampling rate would result in better signal quality in terms of waveform shape. However, the sampling rate of the APS full-frame readout is limited by the power consumption and available area of the system. Since the APS readout records from all 20k electrodes, higher sampling rate would lead to a much higher power consumption in the readout channels (e.g., for the analog-to-digital converters (ADCs)) and a higher output data rate, which would entail issues, such as heating of the chip and difficulties in the acquisition of the data. The low-pass has been set to 3 kHz to limit the noise levels and to get an optimal SNR for reliable spike detection. We added a section in the Discussion that explains the many trade-offs in the design of HD-MEAs. (line 373-383)

For spike sorting, common spike sorting algorithms (kilosort²³, spyking circus²⁴, etc.) also filter the data between 0.3-3 kHz. This has been shown to maintain the characteristic waveform features and information needed to run subsequent spike train analysis. In the DM system, the full-bandwidth is available in the SM mode (24 kHz sampling rate). The DM system can be used to assess waveform shapes using SM data with high sampling rate and high low-pass frequency. However, for many analysis methods demonstrated in the manuscript, such as the spike-triggered averaging and the mapping of network connectivity, only the spike times are needed, so that APS full-frame recording with a sampling rate of 11 kHz is sufficient. We added the relevant information to the manuscript in the discussion section. (line 401-405)

In the initial description of the dual-mode capability, it is important to stress that each electrode can be recorded using both modes simultaneously. This is mentioned in the Methods but not until then.

We have added this information to the manuscript. Also, we have added a schematic as new Figure 1b, which shows the simultaneous operation. (line 94-96)

The heading on line 276- “Mapping dynamics...” – bursting characteristics are not really considered to be a dynamical feature for many neurophysiologists. Different wording would be helpful.

We have changed the heading to “Mapping network activity and putative monosynaptic connections”. (line 306)

Wording issues (minor)

The constructs “enable to” and “allow to” are used throughout the document and this seems incorrect. For instance:

line 60- “enable to simultaneously record from” could be rewritten as “enable simultaneous recording from”

Corrected. (line 74)

line 71- “allows to pack electrodes very close to each other” to “permits dense electrode packing”

Corrected. (line 84)

line 379 “the DM-MEA enables to extract features”, could be changed to “the DM-MEA enables features to be extracted”

Corrected. (line 434)

The sentence in lines 66-70 is rather long. Suggestion: “... addressing scheme within the array: the front-end amplifiers...”

Corrected. (line 80-83)

References

1. Wainger, B. J. *et al.* Intrinsic membrane hyperexcitability of amyotrophic lateral sclerosis patient-derived motor neurons. *Cell Rep.* **7**, 1–11 (2014).
2. Odawara, A., Matsuda, N., Ishibashi, Y., Yokoi, R. & Suzuki, I. Toxicological evaluation of convulsant and anticonvulsant drugs in human induced pluripotent stem cell-derived cortical neuronal networks using an MEA system. *Sci. Rep.* **8**, (2018).
3. Bertotti, G. *et al.* A CMOS-Based Sensor Array for In-Vitro Neural Tissue Interfacing with 4225 Recording Sites and 1024 Stimulation Sites. in *Biomedical Circuits and Systems Conference (BioCAS)* 304–307 (IEEE, 2014). doi:10.1109/BioCAS.2014.6981723.
4. Eversmann, B. *et al.* CMOS Sensor Array for Electrical Imaging of Neuronal Activity. *2005 IEEE Int. Symp. Circuits Syst.* 3479–3482 (2005) doi:10.1109/ISCAS.2005.1465378.
5. Tsai, D., Sawyer, D., Bradd, A., Yuste, R. & Shepard, K. L. A very large-scale microelectrode array for cellular-resolution electrophysiology. *Nat. Commun.* **8**, 1802 (2017).
6. Berdondini, L. *et al.* Active pixel sensor array for high spatio-temporal resolution electrophysiological recordings from single cell to large scale neuronal networks. *Lab Chip* **9**, 2644 (2009).
7. Bakkum, D. J. *et al.* Tracking axonal action potential propagation on a high-density microelectrode array across hundreds of sites. *Nat. Commun.* **4**, 4:2181 (2013).
8. Radivojevic, M. *et al.* Electrical Identification and Selective Microstimulation of Neuronal Compartments Based on Features of Extracellular Action Potentials. *Sci. Rep.* **6**, 31332 (2016).
9. Bakkum, D. J. *et al.* The Axon Initial Segment is the Dominant Contributor to the Neuron’s Extracellular Electrical Potential Landscape. *Adv. Biosyst.* **3**, (2019).

10. Bullmann, T. *et al.* Large-Scale Mapping of Axonal Arbors Using High-Density Microelectrode Arrays. *Front. Cell. Neurosci.* **13**, (2019).
11. Müller, J. *et al.* High-resolution CMOS MEA platform to study neurons at subcellular, cellular, and network levels. *Lab Chip* **15**, 2767–2780 (2015).
12. Abdelfattah, A. S. *et al.* Bright and photostable chemigenetic indicators for extended in vivo voltage imaging. *Science (80-.)*. **365**, 699–704 (2019).
13. Ji, N., Freeman, J. & Smith, S. L. Technologies for imaging neural activity in large volumes. *Nature Neuroscience* vol. 19 1154–1164 (2016).
14. Laissue, P. P. *et al.* Assessing phototoxicity in live fluorescence imaging. *Nature Publishing Group* vol. 14 <http://cbs.umn.edu/cgc/> (2017).
15. Smith, N. A. *et al.* Fluorescent Ca²⁺ indicators directly inhibit the Na,K-ATPase and disrupt cellular functions. *Sci. Signal.* **11**, (2018).
16. Obien, M. E. J., Deligkaris, K., Bullmann, T., Bakkum, D. J. & Frey, U. Revealing neuronal function through microelectrode array recordings. *Front. Neurosci.* **8**, (2015).
17. Dragas, J. *et al.* In Vitro Multi-Functional Microelectrode Array Featuring 59 760 Electrodes, 2048 Electrophysiology Channels, Stimulation, Impedance Measurement, and Neurotransmitter Detection Channels. *IEEE J. Solid-State Circuits* **52**, 1576–1590 (2017).
18. Lopez, C. M. *et al.* A multimodal CMOS MEA for high-throughput intracellular action potential measurements and impedance spectroscopy in drug-screening applications. *IEEE J. Solid-State Circuits* **53**, 3076–3086 (2018).
19. Frey, U. *et al.* Switch-Matrix-Based High-Density Microelectrode Array in CMOS Technology. *Solid-State Circuits, IEEE J.* **45**, 467–482 (2010).
20. Ballini, M. *et al.* A 1024-Channel CMOS Microelectrode Array With 26,400 Electrodes for Recording and Stimulation of Electrogenic Cells In Vitro. *IEEE J. Solid-State Circuits* **49**, 2705–2719 (2014).
21. Radivojevic, M. *et al.* Tracking individual action potentials throughout mammalian axonal arbors. *Elife* **6**, e30198 (2017).
22. Jäckel, D. *et al.* Combination of High-density Microelectrode Array and Patch Clamp Recordings to Enable Studies of Multisynaptic Integration. *Sci. Rep.* **7**, 1–17 (2017).
23. Pachitariu, M., Steinmetz, N., Kadir, S., Carandini, M. & Harris, K. *Fast and accurate spike sorting of high-channel count probes with KiloSort.* (2016).
24. Yger, P. *et al.* A spike sorting toolbox for up to thousands of electrodes validated with ground truth recordings in vitro and in vivo. *Elife* **7**, (2018).

Reviewers' Comments:

Reviewer #1:

Remarks to the Author:

Reviewers Comments; NCOMMS-20-16492A Revised

The authors have addressed all my concerns and the manuscript now reads quite smoothly. I have only a few comments:

Lines 124-125 is not a complete sentence

Line 207 "significantly higher spatiotemporal resolution" compared to wide field voltage imaging. While the temporal resolution is certainly better, and 18 um pixel size have hardly be argued for better spatial resolution. Temporal certainly, but not spatial for this comparison. For prior MEA, certainly.

Finally, I found the discussion to be almost entirely redundant. I certainly didn't learn anything the text and figures had not taught. I would urge the authors to eliminate the discussion and add any critical tidbits in the summary or drastically shorten the discussion.

I do not need to see the manuscript again. The paper is much improved and the community will benefit

Reviewer #2:

Remarks to the Author:

Authors have addressed all the concerns raised by this reviewer.

One last point that should be added:

In Figure 3 (amplitude vs spatial distribution plot, moved from previous Fig 1b), is this the summary of the different preparations authors tried (a,b,c,d,e,f)? If so, please add the description in caption. It would be helpful if authors add (a), (b), ... (f) in the plot beside each prep.

Reviewer #3:

Remarks to the Author:

I have one remaining comment/issue. In the authors' response to the sampling rate issue, it seems that it would be worthwhile discussing some work-arounds that would take advantage of the dual-mode capabilities of their device. Correct isolation of action potential waveforms is critical in certain situations, for instance when looking at propagation of an action potential along a fiber located in a bundle where other action potentials are propagating simultaneously. In this situation it is likely that multiple waveforms will be detected and the identity of a particular unit could be exchanged unless isolation is carefully maintained. In this example, high-resolution, low-noise recording is critical. It would be helpful to discuss how this situation could be handled for instance, when doing spike-triggered averaging using the mixed mode. In this case, the trigger event could be recorded in SM mode while sampling from a bundle of axons with APS mode. SM mode recordings from the

bundle could then be carried out and these waveforms compared to those from STA. If the averages reflect true single-units, then the two waveform set should be quite similar.

Response to reviewers' comments

Reviewer #1

The authors have addressed all my concerns and the manuscript now reads quite smoothly. I have only a few comments:

Lines 124-125 is not a complete sentence

We have modified the sentence at line 124.

Line 207 “significantly higher spatiotemporal resolution” compared to wide field voltage imaging. While the temporal resolution is certainly better, and 18 um pixel size have hardly be argued for better spatial resolution. Temporal certainly, but not spatial for this comparison. For prior MEA, certainly.

We agree with the reviewer and changed the argument to “significantly higher temporal resolution”.

Finally, I found the discussion to be almost entirely redundant. I certainly didn't learn anything the text and figures had not taught. I would urge the authors to eliminate the discussion and add any critical tidbits in the summary or drastically shorten the discussion.

We have shortened the Discussion part and have removed some redundant parts.

I do not need to see the manuscript again. The paper is much improved and the community will benefit

Reviewer #2

In Figure 3 (amplitude vs spatial distribution plot, moved from previous Fig 1b), is this the summary of the different preparations authors tried (a,b,c,d,e,f)? If so, please add the description in caption. It would be helpful if authors add (a), (b), ... (f) in the plot beside each prep.

We thank the reviewer for the suggestion. We have modified Figure 3 accordingly.

Reviewer #3

I have one remaining comment/issue. In the authors' response to the sampling rate issue, it seems that it would be worthwhile discussing some work-arounds that would take advantage of the dual-mode capabilities of their device. Correct isolation of action potential waveforms is critical in certain situations, for instance when looking at propagation of an action potential along a fiber located in a bundle where other action potentials are propagating simultaneously. In this situation it is likely that multiple waveforms will be detected and the identity of a particular unit could be exchanged unless isolation is carefully maintained. In this example, high-resolution, low-noise recording is critical. It would be helpful to discuss how this situation could be handled for instance, when doing spike-triggered averaging using the mixed mode. In this case, the trigger event could be recorded in SM mode while sampling from a bundle of axons with APS mode. SM mode recordings from the bundle could then be carried out and these waveforms compared to those from STA. If the averages reflect true single-units, then the two waveform set should be quite similar.

We thank the reviewer for pointing this out. Indeed we could use the dual-mode capability to verify if the STA works properly. In Supplementary Figure 3, we compared the STA results using DM, APS and SM modes, which showed very similar waveforms for the three modes for the trigger electrode (in grey) as well as for an electrode along the axon (in orange). The same could be attempted across axonal bundles using SM recordings as triggers along the axon, and the waveforms obtained by STA with the APS mode could be compared with those obtained by the SM. STA would yield reliable results if the waveforms would be similar or equal.